# AutoVP: An Automated Visual Prompting Framework and Benchmark

**Hsi-Ai Tsao[1*], Lei Hsiung[2*], Pin-Yu Chen[3], Sijia Liu[4], Tsung-Yi Ho[5]**
[1] National Tsing Hua University, [2] Dartmouth College, [3] IBM Research
[4] Michigan State University, [5] The Chinese University of Hong Kong
[*] Equal contribution

## Abstract

Visual prompting (VP) is an emerging parameter-efficient fine-tuning approach to adapting pre-trained vision models to solve various downstream image-classification tasks. However, there has hitherto been little systematic study of the design space of VP and no clear benchmark for evaluating its performance. To bridge this gap, we propose **AutoVP**, an end-to-end expandable framework for automating VP design choices, along with 12 downstream image-classification tasks that can serve as a holistic VP-performance benchmark. Our design space covers 1) the joint optimization of the prompts; 2) the selection of pre-trained models, including image classifiers and text-image encoders; and 3) model output mapping strategies, including nonparametric and trainable label mapping. Our extensive experimental results show that AutoVP outperforms the best-known current VP methods by a substantial margin, having up to 6.7% improvement in accuracy; and attains a maximum performance increase of 27.5% compared to linear-probing (LP) baseline. AutoVP thus makes a two-fold contribution: serving both as an efficient tool for hyperparameter tuning on VP design choices, and as a comprehensive benchmark that can reasonably be expected to accelerate VP's development. The source code is available at `https://github.com/IBM/AutoVP`.

## 1 Introduction

Originating in the domain of natural language processing, prompting (Gao et al., 2021; Lester et al., 2021; Shi et al., 2023) has gained considerable popularity as a parameter-efficient fine-tuning approach for adapting pre-trained models to downstream tasks. Prompting's methodology has recently been extended to the field of computer vision, where it is termed visual prompting (VP) (Bahng et al., 2022). In its simplest form, VP can be perceived as an in-domain model-reprogramming technique (Chen, 2022). More specifically, it adjusts the inputs and outputs of a pre-trained vision model to address downstream image-classification tasks, without having to make any changes to the weights or architecture of the source model's pre-trained backbone. As such, it stands in contrast to conventional transfer-learning methods that involve complete fine-tuning, LP (i.e., involving modifications of the trainable linear layer in the penultimate layer's output), or zero-shot learning (Radford et al., 2021). For instance, as illustrated in Figure 1, VP adds a universal trainable data frame to the target samples at the model-input stage, and then employs a mapping function – which can be either explicitly defined or implicitly learned – to associate the source and target labels at the output stage.

While VP exhibits tremendous potential, there are two critical challenges that limit its research and development. The first is *absence of a systematic VP framework*. That is, VP design choices, such as prompts' sizes and shapes, source models, and label-mapping (LM) strategies, have thus far only been studied separately, generally for the purpose of delineating their distinct roles in enhancing downstream task accuracy. Ideally, such a systematic framework would automatically search for the best configurations for performance optimization. For example, Bahng et al. (2022) have demonstrated that changing the padding size of visual prompts can yield around 15% variation in final accuracy. It has also been observed that VP is better at augmenting large text-image models, such as CLIP (Radford et al., 2021), than pure vision models like ResNet50 (He et al., 2016). In a study by Chen et al. (2023b), *iterative label mapping* (ILM) during training achieved accuracy up to

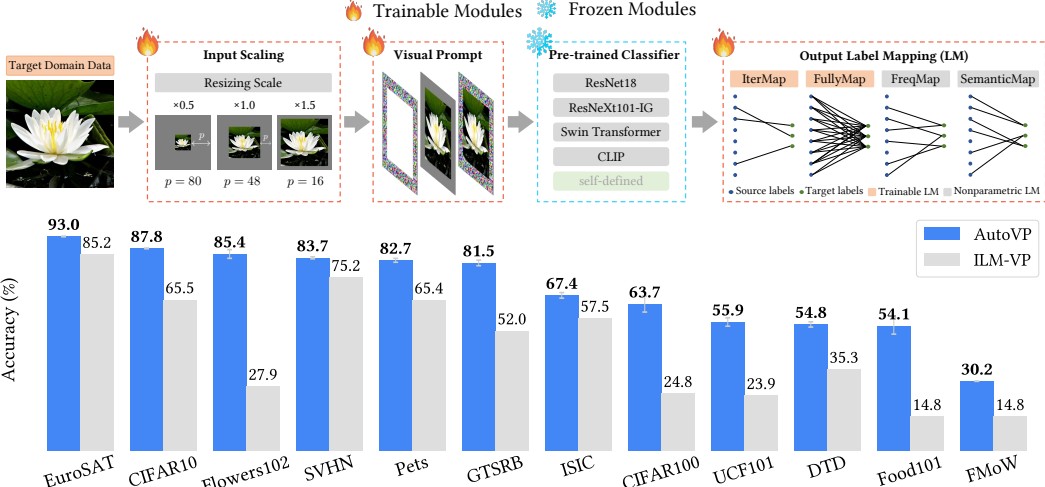

Figure 1: Overview and key highlights of AutoVP. The main components of AutoVP are: **Input Scaling**, which offers three initial input scale options: ×0.5, ×1.0, and ×1.5; **Visual Prompt**, which pads the prompts to the scaled input image; **Pre-trained Classifier**, allowing users (or AutoVP) to select from four pre-trained models: ResNet18 (He et al., 2016), ResNeXt101-IG (Mahajan et al., 2018), Swin-T (Liu et al., 2021), and CLIP (Radford et al., 2021); and **Output Label Mapping**, offering four label mapping options: Iterative Mapping (IterMap), Frequency Mapping (FreqMap), Semantic Mapping (SemanticMap), and Fully Connected Layer Mapping (FullyMap). *Bottom panel*: Given a fixed ImageNet-pre-trained classifier (ResNet18), AutoVP outperforms the state-of-the-art (ILM-VP in Chen et al. (2023b)) on all 12 different downstream image-classification tasks.

13.7% better than fixed label mapping strategies. The second critical challenge is *lack of a unified performance benchmark*: the existing literature evaluates the performance of proposed VP methods in an *ad hoc* manner, by reporting on arbitrarily selected downstream datasets, making comparisons across different methods difficult at best.

To bridge this gap, this paper proposes AutoVP, a solution addressing both these challenges via 1) its automated, extendable framework for joint optimization of a) input-image scaling (i.e., prompt size), b) visual prompts, c) source model selection, and d) output label-mapping strategies; and 2) its provision of a unified benchmark consisting of 12 diverse image-classification tasks with quantifiable content-similarity relative to the dataset (e.g., ImageNet) used for source model pre-training.

As shown in Figure 1, the first component (i.e., input scaling) of AutoVP determines the optimal ratio between the sizes of prompts and the original images. The second, visual prompts, serve as trainable parameters, and undergo iterative updates during the backpropagation phase. The pre-trained model extracts pertinent features and renders predictions within the source domain; and finally, output label mapping establishes a connection between the label spaces of the source and target domains, facilitating accurate predictions in the downstream domain. The modularity of AutoVP allows for the seamless integration and easy extension of various designs for these four components.

Table 1 compares AutoVP against prior VP proposals and the other two baselines proposed to date: LP and text-prompt (TP)-based zero-shot inference. As the table shows, AutoVP is the only such framework that considers the broad range of settings that can affect VP performance. Moreover, thanks to such settings' collective optimization, AutoVP's configuration amounts to a breakthrough in average accuracy across 12 distinct downstream tasks. For instance, with CLIP as the pre-trained model (see Table 2), AutoVP's average accuracy is 4.6% higher than CLIP-VP's (Bahng et al., 2022) and 2.1% higher than ILM-VP's (Chen et al., 2023b). AutoVP also surpasses LP's accuracy by 0.7% on average, suggesting that it is a competitive alternative to LP in terms of transfer learning.

We summarize the main contributions as follows:

- AutoVP is the first end-to-end VP framework that simultaneously takes account of the design of input scale, visual prompts, pre-trained model selection, and output LM strategies. This modular approach to automating VP gives its users flexibility for VP engineering, as well as a straightforward, comprehensive performance benchmark based on 12 downstream image-classification datasets.

Table 1: Comparison of AutoVP with other baselines, including Linear Probing, CLIP zero-shot inference with text prompts (i.e. CLIP-TP in Radford et al. (2021)), CLIP-VP (Bahng et al., 2022), and ILM-VP (Chen et al., 2023b). The average accuracy is evaluated over 12 downstream tasks (see Section 4). For detailed information about the setting configurations, please refer to Section 3.

| Method | Pre-trained Classifier | Prompt Size | Output Transformation | Output Mapping Number | Average Accuracy (%) |
|---|---|---|---|---|---|
| Linear Probing | CLIP | — | Modified Last Classification Layer | — | 79.86 |
| CLIP-TP | CLIP | — | Fixed Text Prompt | 1 | 49.54 |
| CLIP-VP | CLIP | 30 | Fixed Text Prompt | 1 | 76.01 |
| ILM-VP | ResNet18 CLIP | 48 30 | IterMap | 1 | 45.19 78.45 |
| **AutoVP** (Ours) | ResNet18 ResNeXt101-IG Swin-T CLIP | Trainable | IterMap FullyMap FreqMap SemanticMap | 1/5/10 | 81.02 |

- The proposed hyper-parameter tuning process is capable of identifying optimal configurations tailored to individual downstream datasets. In addition, its novel components – e.g., automated input scaling (Section 3) and weight initialization (Section 4.2) – augment VP's overall efficacy significantly, as compared to state-of-the-art VP methods, LP, and zero-shot baselines (see Table 1).

- This paper represents the first step in a comprehensive exploration of optimal configurations across varied conditions (e.g., fixing a source model or an output-mapping strategy), and presents an analysis of domain similarity's impact on VP performance for each downstream dataset.

- This paper highlights AutoVP's superior performance over LP in data-limited settings (Figure 2) and its better out-of-distribution robustness than LP (Figure 4).

## 2 BACKGROUND AND RELATED WORK

**Background of Visual Prompts.** Traditionally, to derive task-specific machine-learning models, researchers have to train or update all model parameters. But, amid the advancement of powerful foundation models, model fine-tuning and training from scratch have both become time-consuming and parameter-inefficient approaches, usually requiring large amounts of training data and storage space. To this end, VP, also known as in-domain model reprogramming, has emerged as an effective means of obtaining machine-learning models for various domain-specific tasks (Chen, 2022). A well-developed pre-trained model from a source domain can be directly used for performing tasks in the target domain with little transformation of the target data. For example, we can use an ImageNet pre-trained model to classify medical images without modifying any of its parameters (Tsai et al., 2020). On the other hand, VP, along with temperature scaling, can also be used as a post-processing calibration method to align model confidence and accuracy (Tang et al., 2022; Hsiung et al., 2023). As compared to traditional approaches such as transfer learning, model fine-tuning, and training from scratch, VP is a low-complexity and model-agnostic strategy; and it is especially suitable for low-data domains.

**The Design of Visual Prompts.** A VP framework can generally be divided into two *trainable* modules, one for *input transformation* and the other for *output transformation*. These are respectively placed at the input and output ends of a pre-trained model. In the case of input transformation, previous literature has proposed various ways to generate and place visual prompts. One of the most popular such approaches is to pad a frame around the target task image and then fill it with trainable additive-input perturbation (prompts) (Tsai et al., 2020; Chen et al., 2023b; Elsayed et al., 2019; Bahng et al., 2022; Wu et al., 2022; Oh et al., 2023). Next, since the output logits of the source pre-trained model are still in the source domain, further output transformation (e.g., LM) is required to obtain the target-domain logits. One naive way of achieving this is randomly mapping (RandMap) $m$ source labels onto the target labels. Tsai et al. (2020) found that frequency-based LM (FreqMap), which constructs its LM from the source-label prediction distribution of the target-domain data, can further improve the accuracy of downstream tasks. However, Chen et al. (2023b) argued that FreqMap lacks interpretability and that its interaction with VP is difficult to measure. To address that

problem, the authors proposed iterative LM (IterMap), a transformation of FreqMap that enables it to concurrently design LM and visual prompts. Yang et al. (2023), meanwhile, proposed a semantics-based LM approach that aligns source and target classes that have similar semantic meanings. And Liao et al. (2023) utilized a prototypical verbalizer to map a mask token to downstream labels, thus providing a different perspective on LM. In this paper, we follow a similar design to Bahng et al. (2022), in which visual prompts are placed around images for input transformations, and there are four mapping methods for output transformations. Further details will be presented in Section 3.

**Non-universal Visual Prompts.** Instead of utilizing universal input prompts, some researchers have focused on designing input-aware prompting models (Zhou et al., 2022a;b). For instance, Chen et al. (2023a) generated class-wise visual prompts to improve model robustness. Similarly, to address accuracy drops caused by low-voltage-induced bit errors, Sun et al. (2023) proposed an input-aware add-on module to generate a robust prompt; and Loedeman et al. (2022) proposed the Prompt Generation Network (PGN), which generates visual prompt token vectors based on input images, allowing for more adaptive and context-aware prompting.

Although input prompting is commonly applied directly to the target image, researchers have also developed other prompting methods, such as *convolutional visual prompt* (Tsai et al., 2023), which learns prompting parameters in a small convolutional structure through self-supervision tasks without knowledge of ground truths, and *visual prompt tuning* (Jia et al., 2022; Sohn et al., 2023), which learns prompting parameters at intermediate layers of a source model. In this paper, we focus on a pixel-level VP approach using a task-specific prompting model for each image-classification task. As such, our approach closely resembles real-world scenarios in which a pre-trained source model remains unmodified, and external variations are not introduced internally.

## 3 AutoVP Framework

Following the system overview of AutoVP in Figure 1, we present its four main components (Input Scaling, Visual Prompt, Pre-trained Classifier, and Output Label Mapping) and its hyper-parameter tuning feature, which enables the joint optimization of these components. Our framework can be extended to support user-defined configurations.

**Input Scaling.** In our implementation of AutoVP, we choose frame-shape prompts as the default prompting template. Hence, the visual prompt sizes $p$ represent the width of the frame, and its actual number of parameters is $2cp(h + w - 2p)$, where $c$, $w$, and $h$ are color channels, width and height respectively. Although the input image size is determined by the source model, when fine-tuning to a downstream dataset from a source model, there is design freedom to resize the target images and use the remaining space for visual prompts. For instance, if the source model takes images with size $224 \times 224$ as input, one can scale the target image size to $128 \times 128$, resulting in the final visual prompt of size $p = (224 - 128)/2 = 48$. It was shown in Bahng et al. (2022) and Wu et al. (2022) that the prompt size ($p$) plays a key role in VP performance. To automate the process of optimizing image resizing scale, we design a gradient-based optimization algorithm to implement the *input scaling module*, which is implemented using `kornia.geometry.transform()` from the Kornia library (Riba et al., 2020). The `transform()` function integrates a range of geometric transformations for 2D images into deep learning, including differentiable image scaling. In addition to image resizing, the prompt size $p$ will also scale up or down to fill the remaining space. Furthermore, to facilitate the optimization of image resizing and avoid bad local minima, we set the default image size to 128 along with three initial scales: 0.5, 1.0, and 1.5 to optimize, and the corresponding prompt sizes $p$ are 80, 48, and 16 respectively. Consequently, the input scaling module allows AutoVP to obtain the optimal image resizing scale and prompt size ($p$).

**Visual Prompt.** For the visual prompt module, AutoVP adds universal pixel-level prompts around all (resized) input images. Let $x_t \in \mathbb{R}^{N_t}$ denote the *target* (flattened) input image (of $N_t$-dimension), $\tilde{x}_t \in \mathbb{R}^{N_s}$ be the prompted image, which fits the input dimension ($N_s$) of the pre-trained source model $f_{\theta_s}$ ($\theta_s$ denotes its weights), $\delta \in \mathbb{R}^{N_s}$ be a trainable universal perturbation, and $\mathcal{M}_p \in \{0, 1\}^{N_s}$ be a binary mask of prompt size $p$, indicating the prompting area. Hence, the prompted image $\tilde{x}_t$ can be formulated as:

$$\tilde{x}_t = \mathcal{P}(x_t) = \text{InputScaling}_p(x_t) + \underbrace{\mathcal{M}_p \odot \sigma(\delta)}_{\text{Prompts}}. \tag{1}$$

The prompts are initialized as 0 and formally defined as $\mathcal{M}_p \odot \sigma(\delta)$, where $\sigma$ is the Sigmoid function that maps the input to a value between 0 and 1 (the scaled input pixel value range), ensuring it has the same numerical range as the input image. We then update $\delta$ using gradient descent.

**Pre-trained Classifier.** After applying the prompts to the resized image through the preceding stages, the prompted image is subsequently fed into the pre-trained model, which serves as a feature extractor to generate predictions in the source domain. We include four representative pre-trained models in our AutoVP framework: ResNet18 (He et al., 2016), ResNeXt101-IG (Mahajan et al., 2018), Swin-T (Liu et al., 2021), and a vision-language multi-modal model, CLIP (Radford et al., 2021) with the ViT-B/32 vision encoder backbone. Note that in AutoVP, the weights of the pre-trained classifiers are frozen and kept unchanged. The details of the models are provided in Appendix A.1.

**Output Label Mapping.** The pre-trained models predict target data to source labels, while the last mile for VP is to map predictions on the source labels to target classes. As illustrated in Figure 1, AutoVP provides four types of output mapping, and they can be generally categorized into two groups. (i) *nonparametric* label mapping: frequency mapping (FreqMap) and semantic mapping (SemanticMap), which are defined during the initialization of VP training and remain unchanged throughout the training process; and (ii) *trainable* label mapping: iterative label mapping (IterMap) and fully connected layer mapping (FullyMap). These two methods dynamically adjust the mapping based on the prompted images. In the following, we provide the overview of our four output mapping approaches, please refer to Appendix A.2 for more details.

- **Frequency Mapping (FreqMap)** is proposed by Tsai et al. (2020). It utilizes the source-label prediction distribution of the target-domain data to map each target class to the top-$m$ most frequent source classes. Let $\mathcal{Y}_s = \{0, \cdots, K_s - 1\}$ and $\mathcal{Y}_t = \{0, \cdots, K_t - 1\}$ be the set of source and target labels, where $K_s/K_t$ are the numbers of source/target classes. Consider $\tilde{\mathcal{X}}_t$ collects all prompted images of label $y_t$ in target domain $\mathcal{D}_t$, i.e. $\tilde{\mathcal{X}}_t = \{\tilde{x}_{t_i} = \mathcal{P}(x_{t_i}) | (y_{t_i} = y_t), (x_{t_i}, y_{t_i}) \in \mathcal{D}_t\}$, then when $m = 1$, the mapping of $y_t$ can be defined as:

$$y_t \leftarrow y_s^* = \arg\max_{y_s \in \mathcal{Y}_s} (\sum_{\tilde{x}_t \in \tilde{\mathcal{X}}_t} Pred(f_{\theta_s}(\tilde{x}_t), y_s)),$$

$$Pred(f_{\theta_s}(\tilde{x}_t), y_s) = \begin{cases} 1, & \text{if } y_s = \arg\max f_{\theta_s}(\tilde{x}_t) \\ 0, & \text{otherwise} \end{cases}.$$

(2)

  The objective of FreqMap is to map the target label $y_t$ to the source label $y_s^*$, which is the most frequent label that $f_{\theta_s}$ classified $\tilde{\mathcal{X}}_t$ as. If a source class is selected as the most frequently predicted class for multiple target classes, it will be assigned to the target class that has the highest count of predictions. The general many-to-one frequency mapping algorithm is provided in Algorithm 1 in the Appendix A.2. Moreover, random label mapping (RandMap) can be viewed as a special case of FreqMap by randomly assigning a subset of source labels to a target label.

- **Iterative Mapping (IterMap, or ILM)** is proposed by Chen et al. (2023b), which is an iterative approach for updating FreqMap. IterMap performs the frequency mapping at the beginning of each training epoch to obtain a new mapping distribution that aligns with the updated prompts.

- **Semantic Mapping (SemanticMap)** follows the works from Yang et al. (2023) and Yen et al. (2021). We utilize the text encoder of CLIP to generate the embeddings of the names of the source and target classes. Subsequently, we map the source-target pairs based on the highest cosine similarity score between their respective embeddings. Hence, SemanticMap can be utilized in any of the three vision pre-trained models (ResNet18, Swin-T, and ResNeXt101-IG) by establishing mappings between the target classes and semantically similar classes from ImageNet-1K. However, SemanticMap is not applicable for CLIP, as it does not have an explicit set of source domain classes.

- **Fully Connected Layer Mapping (FullyMap)** uses a linear layer to map the source output logits to target classes (Arif et al., 2023). FullyMap can be represented as $L_t = w \cdot L_s + b$, where $L_s$ is the output logits from the source pre-trained model, $w$ and $b$ are the weight and bias vector of the linear layer, and $L_t$ is the output of the linear layer which also serves as the final output logits of the VP model.

**End-to-end Hyper-parameter Tuning.** AutoVP's overall tuning procedure is depicted in Appendix A.3. Given its flexibility and modularity, its users must consider numerous settings ($n = 222$),

Table 2: Comparison of VP testing accuracy (%) using CLIP as a pre-trained model on 12 datasets; the optimal tuning settings of AutoVP and the final prompts sizes $p$ are also provided. In the AutoVP setting field, the notation "*Mapping-m*" represents mapping $m$ source classes to each target class.

| Dataset | AutoVP Setting | **AutoVP** | ILM-VP | CLIP-VP | LP |
|---|---|---|---|---|---|
| SVHN (Netzer et al., 2011) | FullyMap, $p = 51$ | $92.9 \pm 0.2$ | 91.2 | 88.4 | 65.4 |
| CIFAR10 (Krizhevsky & Hinton, 2009) | IterMap-1, $p = 23$ | $95.2 \pm 0.0$ | 94.4 | 94.2 | 95.0 |
| Flowers102 (Nilsback & Zisserman, 2008) | FullyMap, $p = 16$ | $90.4 \pm 0.6$ | 83.7 | 70.3 | 96.9 |
| Food101 (Bossard et al., 2014) | FreqMap-1, $p = 16$ | $82.3 \pm 0.1$ | 79.1 | 78.9 | 84.6 |
| UCF101 (Soomro et al., 2012) | FullyMap, $p = 16$ | $73.1 \pm 0.6$ | 70.6 | 66.1 | 83.3 |
| OxfordIIITPet (Parkhi et al., 2012) | FreqMap-10, $p = 16$ | $88.2 \pm 0.2$ | 85.0 | 85.0 | 89.2 |
| CIFAR100 (Krizhevsky & Hinton, 2009) | FullyMap, $p = 31$ | $77.9 \pm 0.6$ | 73.9 | 75.3 | 80.0 |
| EuroSAT (Helber et al., 2019) | FullyMap, $p = 16$ | $96.8 \pm 0.2$ | 96.9 | 96.4 | 95.3 |
| DTD (Cimpoi et al., 2014) | FullyMap, $p = 17$ | $62.5 \pm 0.3$ | 63.9 | 57.1 | 74.6 |
| ISIC (Codella et al., 2019; Tschandl et al., 2018) | IterMap-1, $p = 16$ | $74.0 \pm 1.0$ | 73.3 | 75.1 | 71.9 |
| FMoW (Christie et al., 2018) | FullyMap, $p = 16$ | $40.8 \pm 0.8$ | 36.8 | 32.9 | 36.3 |
| GTSRB (Houben et al., 2013) | FullyMap, $p = 80$ | $93.1 \pm 0.2$ | 92.6 | 92.4 | 85.8 |
| Average Accuracy | | **80.6** | 78.5 | 76.0 | 79.9 |

including how big the initial input image should be, whether to use a trainable resizing module, which pre-trained classifiers to adopt, what output-mapping method to implement, and the number of source labels to map for each target label. To speed up the tuning operation and save computational resources, we use Ray Tune (Liaw et al., 2018) along with an early-stop strategy for terminating poor trails. In our experiments, we employed grid searches to test all configurations. An ASHA scheduler (Li et al., 2018) was used to retain the top-$n$ tasks, and we continued training them while stopping the remaining tasks early. We established experimentally that $n = 2$ top tasks were enough to obtain the optimal setting. When the few-epoch tuning process (training 2-5 epochs with each setting) is complete, we select the setting having the highest testing accuracy and conduct complete training using that setting. By using hyper-parameter tuning, AutoVP can efficiently find the best configuration of VP and lead to significant accuracy improvement in downstream tasks.

## 4 EXPERIMENTS

**Experimental Setup.** We evaluated the performance of AutoVP on 12 downstream datasets (CIFAR10, CIFAR100, ISIC, SVHN, GTSRB, Flowers102, DTD, Food101, EuroSAT, OxfordIIITPet, UCF101, and FMoW), which are common datasets when measuring transfer learning generalization. Detailed descriptions of these datasets are given in Appendix B.1. We repeated each AutoVP experiment in triplicate, utilizing a learning rate of 40 with the SGD optimizer for CLIP, and a learning rate of 0.001 with the Adam optimizer for the other pre-trained models. The results of the baselines (CLIP-VP (Bahng et al., 2022) and ILM-VP (Chen et al., 2023b)) were extracted from the reported accuracies in their respective papers (please refer to Appendix B.2 for more details). Our experiments were performed on NVIDIA GeForce RTX 3090 and are implemented with PyTorch.

### 4.1 EXPERIMENTAL RESULTS

**Comparison of AutoVP and Prior Work.** To ensure that our comparison of AutoVP against previously proposed VP approaches was fair, we fixed its source model but relaxed its other hyperparameter tunings. The results of using CLIP as the source model are presented in Table 2, along with the optimal settings arrived at. We compared AutoVP against LP and two state-of-the-art VP baselines, CLIP-VP and ILM-VP, whose configurations can also be found in Table 1. With the optimal configuration chosen via the tuning process, AutoVP outperformed these other approaches by up to 6.7% on nine of the 12 target datasets. Additionally, AutoVP surpassed the LP baseline on half those datasets, by a maximum of 27.5% in the case of SVHN. AutoVP also obtained the best average accuracy.

We observed that AutoVP employed FullyMap as the output transformation on most datasets. We speculate that this might have been because the linear layer has more parameters and thus allows the achievement of better results. Also, when AutoVP selected initial image scales, it had a tendency to

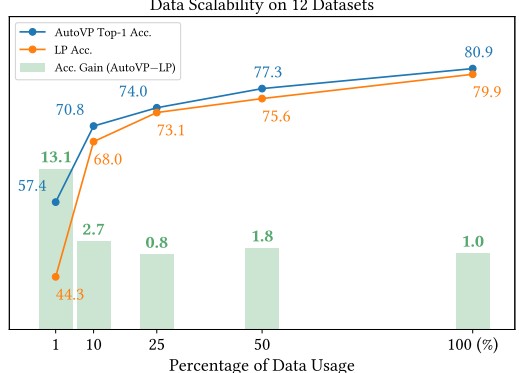
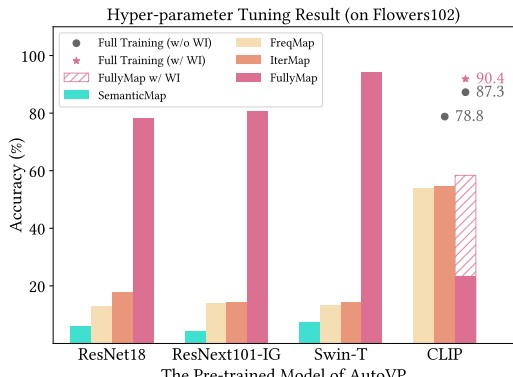

Figure 2: **Data Scalability.** The chart presents the average accuracy of AutoVP and LP across the 12 datasets with varying data percentages: 100%, 50%, 25%, 10%, and 1%. The green bar represents the accuracy gains achieved by AutoVP compared to Linear Probing (LP).

Figure 3: **The Tuning Result on Flowers102 Dataset.** The color bars represent the accuracy tuning for 3 to 5 epochs (with ASHA early-stop optimization). Points illustrate the test accuracy achieved for the best settings given the pre-trained model and mapping method.

scale up those images with relatively small prompt sizes. This allowed the VP model to allocate more attention to the image itself, leading to improved overall performance. As shown in Figure 1, when ResNet18 was used as the source model, AutoVP outperformed ILM-VP by 24.8% on average. More experimental results under this setting are provided in Appendix C.1.

**AutoVP with Source Model Selection.** We also allowed AutoVP to search the optimal source model for downstream tasks. The optimal settings selected by AutoVP, and a comparison of experimental results can be found in Appendix C.1. Our experimental results show that Swin-T was the pre-trained model most frequently chosen by AutoVP as most suitable, i.e., in the cases of eight of the 12 datasets. On average, this choice resulted in 0.43% better accuracy than when CLIP was utilized as the fixed pre-trained backbone. On the DTD and Flowers102 datasets, however, Swin-T's performance was better than CLIP's by much more: i.e., 6.80% and 3.08%, respectively. These findings highlight how multiple pre-trained models can be leveraged to enhance performance across a diverse range of datasets.

**Data Scalability.** To understand how AutoVP would perform in a data-limited regime, we gradually and uniformly reduced the amount of training data to 50%, then 25%, then 10%, and finally 1% of each training dataset's original size. The experimental results in Figure 2 indicate that AutoVP consistently outperformed LP across all 12 datasets, and that its relative performance was especially high in the two scenarios with the lowest data volumes, i.e., 10% and 1% data usage. The dataset-specific results can be found in Figure C.1 (within Appendix C.2).

## 4.2 ABLATION STUDIES OF AUTOVP

We designed a range of model architectures as testbeds for examining the performance of AutoVP's various components. Our comparisons of these VP architectures included 1) the utilization of a weight-initialization strategy with FullyMap, 2) the inclusion *vs.* exclusion of the CLIP text encoder, 3) the presence *vs.* absence of visual prompts, and 4) the frequency analysis of the learned VP.

**Weight Initialization of FullyMap with CLIP.** When CLIP was used as the pre-trained model, the FullyMap output transformation exhibited significantly inferior performance to FreqMap and IterMap (Figure 3). This is because FreqMap and IterMap can leverage CLIP's zero-shot property with the semantic meanings of the labels, whereas the fully connected layer needs to learn the mapping from a random state. As a result, FullyMap tends to perform poorly in the hyper-parameter tuning process, yet may achieve higher accuracy after completing 200 epochs of training. In Figure 3, for example, AutoVP suggests that the optimal output transformation for Flowers102 with CLIP is IterMap; but in reality, FullyMap achieves better performance after training for 200 epochs (87.3%, as against 78.8% for IterMap).

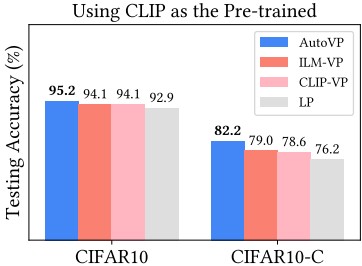

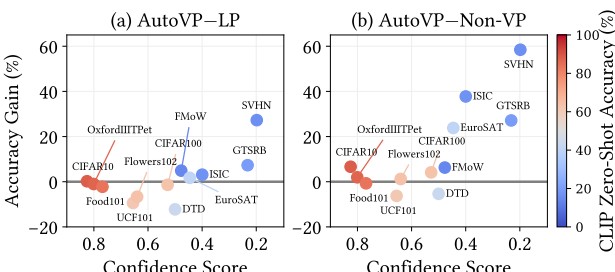

Figure 4: **Models Robustness with CLIP.** The comparison of accuracy drop on the CIFAR10-C dataset across AutoVP, ILM-VP, CLIP-VP and LP.

Figure 5: **Accuracy Gains with CLIP.** The right side of the chart indicates a higher out-of-distribution (OOD) extent, accompanied by larger gain values. Conversely, the left side shows lower gain values.

To address the bottleneck in hyperparameter tuning caused by FullyMap, we introduced weight initialization (WI). This allows FullyMap to initialize with a more informative mapping based on the semantic meaning of the class names. As mentioned in Section 3, AutoVP's FullyMap consists of a linear layer, which can be characterized as $L_t = w \cdot L_s + b$ where the weight $w$ is a $K_t$ by $K_t T$ real matrix, $K_t$ denotes the number of target classes, and $T$ is the number of templates used in the CLIP text encoder. The weight initialization is to assign the diagonal of $w$ to be 1, and the rest of it is set to 0, resulting in $w = (\boldsymbol{I}_{K_t} | \boldsymbol{0})$, where $\boldsymbol{I}_{K_t}$ is an identity matrix of size $K_t$, and $\boldsymbol{0}$ is a $K_t \times K_t(T-1)$ zero matrix, indicating the connection between each target class `class_name` and its corresponding text prompts ("This is a photo of a [`class_name`]"). In Figure A.1 (within Appendix A.2), we represent this concept visually. As shown in Figure 3, our experimental results demonstrate that when utilizing CLIP as the pretrained model, FullyMap with proper WI (hatched bar) can also outperform other mapping approaches.

**Impact of the Non-inclusion of Text Encoder in CLIP.** When replicating the experimental setting as shown in Table 2 and establishing a direct connection between the fully connected output mapping layer and the CLIP image encoder without incorporating the text encoder, there was a substantial decrease in average accuracy: to 69.0% (see Figure E.1, Appendix E.1). Dataset-specific accuracy drops were particularly prominent in Flowers102, Food101, and OxfordIIITPet. These outcomes suggest that label semantics play a crucial role in those datasets.

**The Impact of Visual Prompts.** We also investigated the effects on the overall performance of removing the module of visual prompts from the AutoVP pipeline while retaining the pre-trained model and output-mapping modules. When the ResNet18 model was used, leaving the visual prompts in yielded better performance than omitting them in just three out of 12 cases: i.e., the SVHN, GTSRB, and ISIC datasets (Figure C.3 (b), Appendix C.3). For the remaining datasets, the inclusion of visual prompts actually led to a decline in performance. This suggests that when a relatively small source model is used for VP, a significant improvement in accuracy of the sort reported in Table C.1 can primarily be attributed to fully connected layer mapping, and visual prompts may be perceived as noise. On the other hand, when the CLIP model was used, most of the datasets had positive gain values (Figure 5 (b)), indicating improved performance, when visual prompts were included. This suggests that CLIP is more suitable for VP tasks than ResNet18 is.

**Frequency Analysis of the Learned Visual Prompts.** In Appendix D.1, we also conducted an analysis from a frequency perspective (Brigham, 1988) to study the generalization of visual prompt patterns. The results highlighted the effectiveness of prompting with CLIP, harnessing low-frequency features that generalize to the target domain.

## 5 DISCUSSIONS

**Tuning Selection.** AutoVP provides joint optimization of its multiple configurations and selects different parameters according to its target tasks. In terms of output label mapping, FullyMap exhibits superior performance in vision models, but IterMap or FreqMap appear to enhance the performance of text-image models like CLIP. In this context, weight initialization with FullyMap plays an important role in CLIP, making this option one of the more frequently chosen output-mapping strategies (Table 2). We also observed that novel designs exploiting larger image scales and mapping a larger number

of source classes tended to yield enhanced performance. More information on selection preferences during hyperparameter tuning can be found in Appendix D.3.

**AutoVP Robustness.**   We trained AutoVP on CIFAR10 and evaluated its robustness on the corrupted dataset CIFAR10-C, which consists of 18 types of filters or noise. As shown in Figure 4, AutoVP maintained greater robustness than ILM-VP, CLIP-VP, and LP. Its loss of accuracy was relatively small: suggesting that AutoVP exhibits a lower degree of overfitting to the training data and possesses a higher ability to resist the impact of noise than the other baselines do.

**Performance Evaluation on ID/OOD Downstream Tasks.**   We evaluate the out-of-distribution (OOD) extent of each dataset relative to the pre-trained CLIP by considering the average confidence score (Guo et al., 2017) and the CLIP zero-shot inference. The accuracy gains achieved through VP (Figure 5) were computed as the difference in accuracy between AutoVP and LP or non-VP approaches (i.e. visual prompts were removed and output mapping retained). We observed that the datasets that were more in-distribution (ID), with higher confidence scores and higher zero-shot accuracy, exhibited smaller accuracy gains from VP. Conversely, datasets that were more OOD, characterized by lower confidence scores and lower zero-shot accuracy, had their accuracy improved more through AutoVP.

We also evaluated accuracy gains with ResNet18 pre-trained on ImageNet-1K (Russakovsky et al., 2015) (Figure C.3, Appendix C.3) and, to assess domain differences between ImageNet-1K and other downstream datasets, we calculated the FID score (Heusel et al., 2017). The results were consistent with the cases using CLIP. In conclusion, AutoVP is suitable for datasets that exhibit more OOD characteristics than the source domain dataset.

## 6   LIMITATIONS

This work is subject to some limitations. First, our optimization process did not include certain hyperparameters, notably learning rate and weight decay. This omission stemmed from our primary focus on identifying the best configurations for VP training, and because including such hyperparameters would have greatly increased execution workload. In addition, when it comes to choosing the best pre-trained model to fine-tune on the target dataset, You et al. (2021) also argued that, in general, the sophisticated fine-tuning techniques (e.g., regularization) would not change the ranking of pre-trained models in downstream tasks. Nonetheless, we conducted supplementary tuning experiments encompassing various learning rates and weight decays, the results of which can be found in Table E.3 (within Appendix E.3). In practice, we suggest enabling the tuning of learning rates, weight decay, and other model-specific parameters after the initial hyperparameter tuning phase of AutoVP. The tuning of fundamental hyperparameters could potentially be accelerated with recent advancements in utilizing generalization metrics to identify optimal hyperparameter configurations (Zhou et al., 2023a;b), a subject to be explored in future research.

Another limitation pertains to the scope of AutoVP, which is oriented toward classification tasks. However, we have extended its application to segmentation and detection tasks, as detailed in Appendix E.2. Furthermore, recent studies have demonstrated the success of visual prompts in generative tasks (Ramesh et al., 2021; 2022; Liu & Chilton, 2022; Bar et al., 2022; Sohn et al., 2023). Nevertheless, expanding support for generative tasks will require accommodation of their distinctive requirements, e.g., via the integration of a suitable pre-trained generative model, such as GANs (Goodfellow et al., 2014), VAEs (Kingma & Welling, 2013), or diffusion models (Ho et al., 2020), along with tailored prompt design. Certainly, our results imply that there are many avenues for VP research that merit further exploration.

## 7   CONCLUSION

This paper has introduced AutoVP, an end-to-end framework that automatically selects the optimal VP configuration for a downstream dataset. AutoVP demonstrates superior performance over other state-of-the-art VP methods and transfers learning baselines in both standard and sample-reduced fine-tuning settings. This research has also yielded important insights into optimal VP configurations, the effects of downstream data characteristics on VP, and how robustness against image corruption might be improved. In short, we believe AutoVP is an efficient and expandable tool, and perhaps more importantly, a useful benchmark that will accelerate the development of VP research and technology.

ACKNOWLEDGMENTS

The research described in this paper was conducted in the JC STEM Lab of Intelligent Design Automation, which is funded by The Hong Kong Jockey Club Charities Trust.

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

# APPENDIX

## A    IMPLEMENTATION DETAILS OF AUTOVP

### A.1    PRE-TRAINED CLASSIFIER DETAILS

Our AutoVP framework includes four pre-trained models: ResNet18 (He et al., 2016), ResNeXt101-IG (Mahajan et al., 2018), Swin-T (Liu et al., 2021), and CLIP (Radford et al., 2021). Both ResNet18 and Swin-T models were trained on the ImageNet-1K (Russakovsky et al., 2015) dataset, while ResNeXt101-IG was pre-trained on a large collection of Instagram photos. Additionally, CLIP was trained on a dataset consisting of 400 million pairs of images and corresponding text from the internet.

In the structural differences, the first three models are single-modality vision models. ResNet18 is a typical and relatively small convolutional neural network with residual blocks. ResNeXt101-IG is a deeper residual network that incorporates *cardinality*, which refers to the size of the set of transformations (Xie et al., 2017). Swin-T is a vision transformer that operates by dividing the input image into patches and processing them using the transformer architecture. The last model, CLIP, is a vision-language multi-modal model that calculates the cosine similarity between image embeddings and label text embeddings. The prediction is to select the class with the highest similarity score to the embedding of the input image. The prediction flow is illustrated in Eq. 3, where the input image is denoted as $x$, and $K_t$ represents the size of the predictable class set. The token vector of the $i$-th class label text, obtained from a tokenizer, is denoted as $\boldsymbol{ClsTk_i}$. CLIP utilizes the Image-Encoder() and Text-Encoder() components to extract features from images and text. The resulting image and text embeddings are represented as $\boldsymbol{x_{emb}}$ and $\boldsymbol{t_{emb_i}}$, respectively. The cosine similarity between any pair of (image, text) embeddings can be computed, and the class with the highest cosine similarity is considered as the predicted class $y_{\text{pred}}$.

$$
\begin{aligned}
\boldsymbol{x_{emb}} &= \text{Image-Encoder}(x) \\
\boldsymbol{t_{emb_i}} &= \text{Text-Encoder}(\boldsymbol{ClsTk_i}),\ 0 \leq i < K_t \\
y_{\text{pred}} &= \arg \max_{0 \leq i < K_t} \left( \frac{\boldsymbol{x_{emb}} \cdot \boldsymbol{t_{emb_i}}}{\parallel \boldsymbol{x_{emb}} \parallel \parallel \boldsymbol{t_{emb_i}} \parallel} \right)
\end{aligned}
\tag{3}
$$

By training with pairs of image and text annotations, CLIP is able to learn correlations between visual and textual information, achieving state-of-the-art zero-shot accuracy.

### A.2    OUTPUT LABEL MAPPINGS OF AUTOVP

As mentioned in Section 3, AutoVP incorporates four output label mappings: frequency mapping (FreqMap), iterative mapping (IterMap), semantic mapping (SemanticMap), and fully connected layer mapping (FullyMap). In the following, we provide more details of each mapping algorithm.

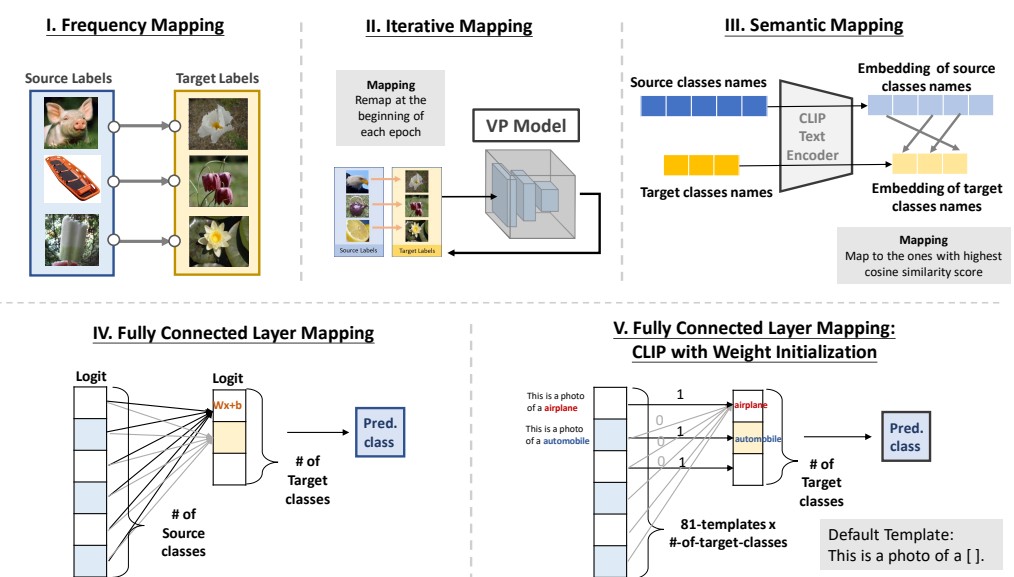

Figure A.1: **Output Label Mapping.** Illustration of four output mapping methods.

### A.2.1 FREQMAP (FREQUENCY MAPPING)

In **FreqMap**, the mapping is established between each target class and the most frequently mapped source class. We demonstrate the general many-to-one mapping (multiple source labels mapped to one target label) in Algorithm 1. In lines 6-10, we traverse all the training data pairs $(x_t, y_t)$. First, we pad the image $x_t$ with the current visual prompts to obtain a prompted image $\tilde{x}_t$. Then, we could obtain the prediction $f_{\theta_s}(\tilde{x}_t) = y_s$ in the source domain. This gives us a mapping relation from $y_s$ to $y_t$, and we increase the count accordingly. In line 11, we obtain the list of source-target id pairs $(S_{id}, T_{id})$, which is sorted by the count matrix (count) in descending order. Accordingly, we start defining the mapping of FreqMap from the most frequently source-target pair. If the source class has not been mapped yet and the mapping count for the target class has not reached the limit $m$, we establish the mapping relationship $\mathbf{M}[S_{id}][T_{id}] = 1$ (line 14). We continue this process until all target classes have been mapped to $m$ source classes. Once this condition is met, the mapping assignment is completed, and we return the mapping matrix $\mathbf{M}$.

---

**Algorithm 1:** FreqMap $(\delta, f_{\theta_s}, \mathcal{D}_t, m)$

**Input:** visual prompts $\delta$, source classifier $f_{\theta_s}$, target dataset $\mathcal{D}_t$, and the specified number of source classes mapped to each target class $m$
**Output:** mapping matrix $\mathbf{M}$

**# Initialization**
1  $K_s,\ K_t \leftarrow$ number of source and target classes
2  $\mathbf{M} \leftarrow \mathbf{0}^{\mathbf{K}_s \times \mathbf{K}_t}$    *# mapping matrix*
3  count $\leftarrow \mathbf{0}^{K_s \times K_t}$     *# a zero matrix, records the number of images of each target class being predicted (by $f_{\theta_s}$) as each source class*
4  done_s $\leftarrow \mathbf{0}^{K_s}$     *# a boolean vector, records whether the source class has been assigned*
5  done_t $\leftarrow \mathbf{0}^{K_t}$     *# a boolean vector, records whether the number of source classes per target class is equal to $m$*

**# Calculate the Frequency**
6  **foreach** $(x_t, y_t) \in \mathcal{D}_t$ **do**
7  $\quad \tilde{x}_t \leftarrow \mathcal{P}(x_t)$     *# generates $\tilde{x}_t$ from $x_t$ and $\delta$ using Eq. 1*
8  $\quad y_s \leftarrow f_{\theta_s}(\tilde{x}_t)$     *# the predicted source domain class*
9  $\quad$ count$[y_s][y_t] \leftarrow$ count$[y_s][y_t] + 1$
10 **end**
11 index_list $\leftarrow$ ArgSort(count)     *# get the list of source-target id pairs $(S_{id}, T_{id})$ sorted by count$[S_{id}][T_{id}]$ in descending order*

**# Define the FreqMap**
12 **for** $(S_{id}, T_{id})$ **in** index_list **do**
13 $\quad$ **if not** done_t$[T_{id}]$ **and not** done_s$[S_{id}]$ **then**
14 $\quad\quad$ $\mathbf{M}[S_{id}][T_{id}] \leftarrow 1$     *# assign the mapping from $S_{id}$ (source label) to $T_{id}$ (target label)*
15 $\quad\quad$ done_s$[S_{id}] \leftarrow$ **True**
16 $\quad$ **end**
17 $\quad$ **if** $Sum(\mathbf{M}[:][T_{id}]) == m$ **then**
18 $\quad\quad$ done_t$[T_{id}] \leftarrow$ **True**     *# if the target class $T_{id}$ has been mapped to $m$ source classes*
19 $\quad$ **end**
20 $\quad$ **if** $Sum($done_t$) == K_t$ **then**
21 $\quad\quad$ **break**     *# early stop if all the target classes has been mapped to $m$ source classes*
22 $\quad$ **end**
23 **end**
24 **return M**

---

### A.2.2 ITERMAP (ITERATIVE MAPPING)

In **IterMap**, at the beginning of each training epoch, the mapping relationship is established using the FreqMap with the current prompted image. This means that the visual prompts updated via the VP model are padded onto the image and participate in the mapping process. Hence, this mapping relationship changes as the visual prompts are trained, which is known as bi-level optimization (Chen et al., 2023b).

### A.2.3 SEMANTICMAP (SEMANTIC MAPPING)

In **SemanticMap**, source and target classes with semantically similar class names are mapped together. This mapping is accomplished using CLIP's tokenizer Tokenizer() and text encoder Text-Encoder(). We demonstrate the mapping process using the following equation:

$$
\begin{aligned}
\boldsymbol{ClsTk_{y_{\mathrm{s}}}} &= \text{Tokenizer}(\boldsymbol{N_{y_{\mathrm{s}}}}) \\
\boldsymbol{ClsTk_{y_{\mathrm{t}}}} &= \text{Tokenizer}(\boldsymbol{N_{y_{\mathrm{t}}}}) \\
\boldsymbol{Emb_{y_{\mathrm{s}}}} &= \text{Text-Encoder}(\boldsymbol{ClsTk_{y_{\mathrm{s}}}}) \\
\boldsymbol{Emb_{y_{\mathrm{t}}}} &= \text{Text-Encoder}(\boldsymbol{ClsTk_{y_{\mathrm{t}}}}) \\
Similarity_{(\boldsymbol{y_{\mathrm{s}}},\boldsymbol{y_{\mathrm{t}}})} &= \frac{\boldsymbol{Emb_{y_{\mathrm{s}}}} \cdot \boldsymbol{Emb_{y_{\mathrm{t}}}}}{\parallel \boldsymbol{Emb_{y_{\mathrm{s}}}} \parallel \parallel \boldsymbol{Emb_{y_{\mathrm{t}}}} \parallel} \\
\boldsymbol{y_{\mathrm{t}}} \leftarrow \boldsymbol{y_{\mathrm{s}}^*} &= \arg \max_{y_{\mathrm{s}} \in \mathcal{Y}_{\mathrm{s}}}(Similarity_{(\boldsymbol{y_{\mathrm{s}}},\boldsymbol{y_{\mathrm{t}}})})
\end{aligned}
\tag{4}
$$

In Eq. 4, let $\mathcal{Y}_{\mathrm{s}} = \{0, \cdots, K_{\mathrm{s}} - 1\}$ and $\mathcal{Y}_{\mathrm{t}} = \{0, \cdots, K_{\mathrm{t}} - 1\}$ be the set of source and target labels, where $K_{\mathrm{s}}/K_{\mathrm{t}}$ are the numbers of source/target classes. For the source label $\boldsymbol{y_{\mathrm{s}}} \in \mathcal{Y}_{\mathrm{s}}$ with the classname $\boldsymbol{N_{y_{\mathrm{s}}}}$ and the target label $\boldsymbol{y_{\mathrm{t}}} \in \mathcal{Y}_{\mathrm{t}}$ with the classname $\boldsymbol{N_{y_{\mathrm{t}}}}$, we first utilize the tokenizer to obtain token vectors ($\boldsymbol{ClsTk_{y_{\mathrm{s}}}}$ and $\boldsymbol{ClsTk_{y_{\mathrm{t}}}}$) corresponding to $\boldsymbol{N_{y_{\mathrm{s}}}}$ and $\boldsymbol{N_{y_{\mathrm{t}}}}$. Then, the text encoder is used to obtain their embeddings ($\boldsymbol{Emb_{y_{\mathrm{s}}}}$ and $\boldsymbol{Emb_{y_{\mathrm{t}}}}$). Pair-wise cosine similarity is calculated between each source and target embeddings, and each target label is mapped to the source label with the highest similarity.

### A.2.4 FULLYMAP (FULLY CONNECTED LAYER MAPPING)

**FullyMap** utilizes a linear layer $L_{\mathrm{t}} = w \cdot L_{\mathrm{s}} + b$ with weights $w$ and bias $b$ to learn the mapping, enabling the transformation of the output source logits $L_{\mathrm{s}}$ to target logits $L_{\mathrm{t}}$. As for weight initialization (WI), it is employed in the case of CLIP with FullyMap. This technique involves setting the weights between the target labels and their default templates as 1, while setting the rest to 0. With WI, the linear layer can achieve a favorable initial mapping state, thereby expediting the process of obtaining a good mapping relation.

### A.3 AUTOVP TUNING PROCESS

In Figure A.2, there are three stages involved in the tuning process, while the **Visual Prompt** component depicted in Figure 1 is not involved in the tuning process, as it does not contain any hyper-parameters. During the **Input Scaling** stage, the initial scale of the input image is determined, and users can choose whether to learn the resize scale during training. In the **Pre-trained Classifier** stage, users have the option to select from four pre-trained models to serve as the feature extractor. The **Output Label Mapping** stage offers four mapping methods to choose from. For FreqMap, IterMap, and SemanticMap, users can specify the number of source classes that are mapped to a single target class.

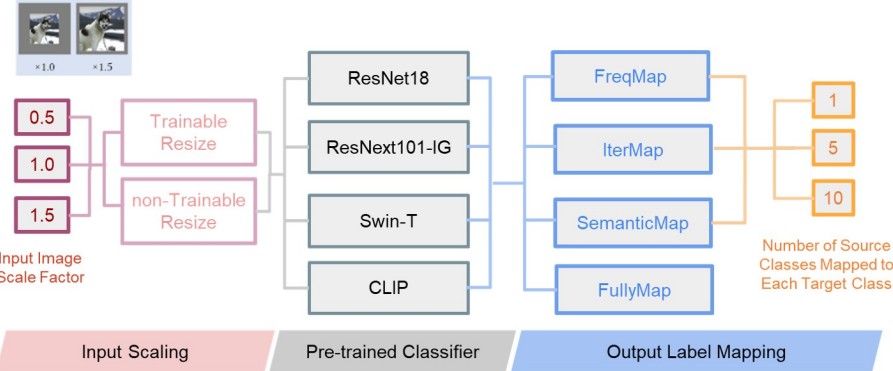

Figure A.2: **Hyper-Parameter Tuning Selection.** Illustration of the end-to-end hyper-parameter tuning process in AutoVP with a total of 222 possible configurations.

# B DATASETS AND BASELINES

## B.1 THE TWELVE DOWNSTREAM DATASETS

To assess the efficacy of the proposed AutoVP, we selected the following twelve datasets for our experiments. The detailed information is shown in B.1.

- **CIFAR10** & **CIFAR100** (Krizhevsky & Hinton, 2009): The datasets consist of labeled subsets of the 80 million tiny images dataset, which are composed of 32×32 color images.
- **ISIC** (Codella et al., 2019; Tschandl et al., 2018): The International Skin Imaging Collaboration (ISIC) developed an international repository of dermoscopic images known as the ISIC Archive. The images of the datasets were acquired using different devices at several medical centers worldwide.
- **SVHN** (Netzer et al., 2011): A real-world image dataset of street view house numbers.
- **GTSRB** (Houben et al., 2013): The German Traffic Sign Benchmark (GTSB) is a multi-class, single-image classification challenge that was conducted at the International Joint Conference on Neural Networks (IJCNN) in 2011.
- **Flowers102** (Nilsback & Zisserman, 2008): The 102 categories flowers dataset consists of commonly occurring flowers in the United Kingdom.
- **DTD** (Cimpoi et al., 2014): The Describable Textures Dataset (DTD) is a collection of textural images that have been annotated with a series of human-centric attributes.
- **Food101** (Bossard et al., 2014): The food dataset consists of 101 different classes, with a total of 101,000 images.
- **EuroSAT** (Helber et al., 2019): The Sentinel-2 satellite images dataset for land use and land cover classification.
- **OxfordIIITPet** (Pets) (Parkhi et al., 2012): The dataset includes diverse breeds of cats and dogs, with images exhibiting variations in scale, pose, and lighting conditions.
- **UCF101** (Soomro et al., 2012): The action recognition dataset consists of realistic action videos that have been collected from YouTube.
- **FMoW** (Christie et al., 2018): The dataset contains satellite images that are used for sites and land use classification.

Table B.1: Dataset Setting

| Dataset | Class Number | Training Set Size | Testing Set Size | Batch Size |
|---|---|---|---|---|
| SVHN | 10 | 73,257 | 26,032 | 128 |
| EuroSAT | 10 | 13,500 | 8,100 | 128 |
| Flowers102 | 102 | 4,093 | 2,463 | 64 |
| CIFAR100 | 100 | 50,000 | 10,000 | 128 |
| UCF101 | 101 | 7,639 | 3,783 | 128 |
| DTD | 47 | 2,820 | 1,692 | 32 |
| FMoW | 62 | 76,863 | 22,108 | 128 |
| GTSRB | 43 | 26,640 | 12,630 | 128 |
| CIFAR10 | 10 | 50,000 | 10,000 | 128 |
| Food101 | 101 | 75,750 | 25,250 | 128 |
| OxfordIIITPet | 37 | 3,680 | 3,669 | 128 |
| ISIC | 7 | 4,990 | 555 | 128 |

## B.2 BASELINES DETAILS

The reported accuracy of the baselines (ILM-VP by Chen et al. (2023b), CLIP-VP and LP by Bahng et al. (2022)) in Section 4 is obtained from the results documented in the respective papers. For some datasets (such as ISIC, FMoW, and GTSRB), the authors did not include them in their paper; in this regard, we follow the corresponding experimental settings to obtain baseline accuracy.

## C  ADDITIONAL EXPERIMENTS WITH AUTOVP

### C.1  FIXED PRE-TRAINED MODEL *vs.* AUTO PRE-TRAINED MODEL SELECTION

We present additional experimental results to highlight the effectiveness of AutoVP. In addition to the comparisons with CLIP and VP baselines discussed in Section 4.1, we further evaluate the performance using ResNet18 as the pre-trained model and explore a scenario without any pre-trained model restrictions. The results using ResNet18 are presented in Table C.1, while the results for the unrestricted scenario are provided in Table C.2. These comparisons consistently demonstrate that AutoVP outperforms previous approaches, including LP and the state-of-the-art VP methods.

Table C.1: **AutoVP with ResNet18.** Comparison of VP test accuracy (%) using ResNet18 as the pre-trained model on 12 datasets.

| Dataset | AutoVP Setting | **AutoVP** | ILM-VP | LP |
|---------|----------------|------------|--------|-----|
| SVHN | FullyMap, $p = 48$ | $83.74 \pm 0.45$ | 75.2 | 65.0 |
| CIFAR10 | FullyMap, $p = 48$ | $87.81 \pm 0.17$ | 65.5 | 85.9 |
| Flowers102 | FullyMap, $p = 16$ | $85.40 \pm 1.89$ | 27.9 | 88.0 |
| Food101 | FullyMap, $p = 16$ | $54.15 \pm 3.53$ | 14.8 | 50.6 |
| UCF101 | FullyMap, $p = 16$ | $55.86 \pm 1.81$ | 23.9 | 63.2 |
| OxfordIIITPet | FullyMap, $p = 16$ | $82.65 \pm 0.84$ | 65.4 | 87.2 |
| CIFAR100 | FullyMap, $p = 16$ | $63.67 \pm 3.48$ | 24.8 | 63.3 |
| EuroSAT | FullyMap, $p = 48$ | $93.01 \pm 0.15$ | 85.2 | 93.8 |
| DTD | FullyMap, $p = 16$ | $54.82 \pm 1.14$ | 35.3 | 60.0 |
| ISIC | FullyMap, $p = 16$ | $67.44 \pm 1.22$ | 57.5 | 68.6 |
| FMoW | FullyMap, $p = 16$ | $30.17 \pm 0.06$ | 14.8 | 28.4 |
| GTSRB | FullyMap, $p = 16$ | $81.52 \pm 1.21$ | 52.0 | 77.4 |
| Average Accuracy | | **70.02** | 45.2 | 69.3 |

Table C.2: **AutoVP with source model selection.** This table displays the best tuning setting without any restriction on the choice of pre-trained model, and shows the test accuracy (%) of AutoVP and the LP baseline of the chosen model across 12 datasets.

| Dataset | AutoVP Setting | **AutoVP** | LP |
|---------|----------------|------------|-----|
| SVHN | CLIP, FullyMap, $p = 51$ | $92.86 \pm 0.18$ | 65.40 |
| CIFAR10 | ResNeXt101-IG, FullyMap, $p = 48$ | $95.89 \pm 0.07$ | 93.89 |
| Flowers102 | Swin-T, FullyMap, $p = 16$ | $93.48 \pm 0.52$ | 95.75 |
| Food101 | CLIP, FreqMap-1, $p = 16$ | $82.28 \pm 0.09$ | 84.60 |
| UCF101 | Swin-T, FullyMap, $p = 16$ | $72.96 \pm 0.26$ | 75.96 |
| OxfordIIITPet | Swin-T, FullyMap, $p = 16$ | $90.20 \pm 0.55$ | 93.04 |
| CIFAR100 | ResNeXt101-IG, FullyMap, $p = 48$ | $79.76 \pm 0.47$ | 76.09 |
| EuroSAT | Swin-T, FullyMap, $p = 16$ | $95.98 \pm 0.02$ | 95.50 |
| DTD | Swin-T, FullyMap, $p = 16$ | $69.25 \pm 0.58$ | 71.49 |
| ISIC | Swin-T, FullyMap, $p = 16$ | $71.66 \pm 1.45$ | 72.22 |
| FMoW | Swin-T, FullyMap, $p = 48$ | $39.79 \pm 0.83$ | 32.73 |
| GTSRB | Swin-T, FullyMap, $p = 55$ | $88.10 \pm 2.11$ | 74.97 |
| Average Accuracy | | **81.02** | 77.64 |

## C.2 Data Scalability

In the data scalability experiments, Figure C.1 illustrates the performance of AutoVP and LP on each dataset under various data usage proportions. The corresponding settings can be found in Table C.3. When the chosen pre-trained model is fixed to CLIP, AutoVP outperformed LP in scenarios with limited data on most of the datasets. In some datasets, such as SVHN, GTSRB, and EuroSAT, AutoVP performs better than LP for all proportions of data. Besides, in OxfordIIITPet, CIFAR100, DTD, Food101, UCF101, and Flowers102, AutoVP also obtains promising performance compared to LP on a small amount of data. This suggests that AutoVP will be a more suitable and effective solution than LP, especially when training data is limited.

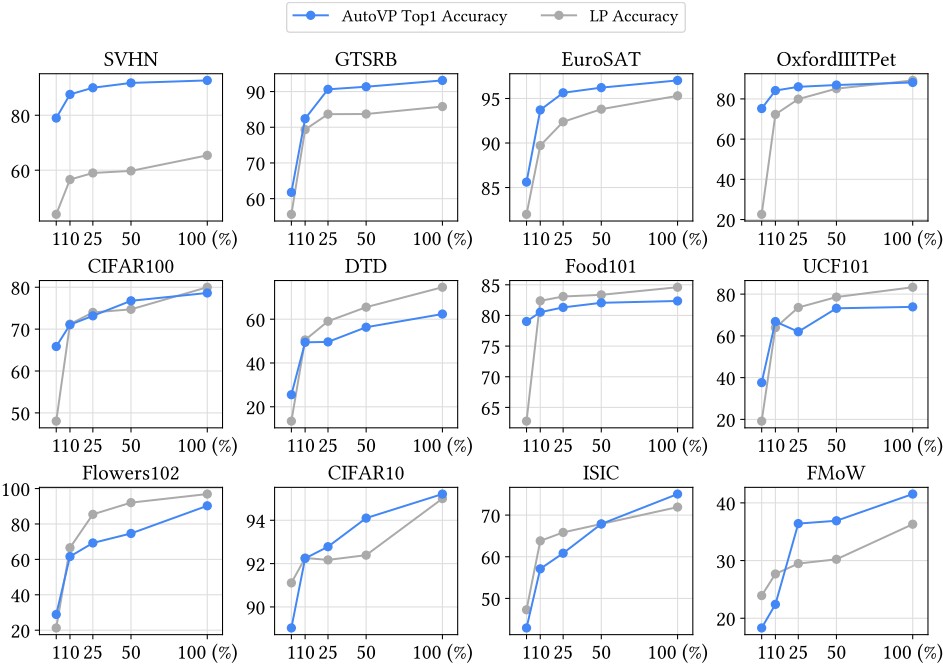

Figure C.1: **Data Scalability.** The charts present the accuracy of AutoVP and LP with varying percentages of data usage: 100%, 50%, 25%, 10%, and 1%.

Table C.3: **Data Scalability Settings.** The table shows the settings for data scalability experiments with data usage ranging from 100% to 1%. The source pre-trained model is fixed to CLIP. The notation "*Mapping-m*" represents mapping $m$ source classes to each target class.

| Dataset | 100% | 50% | 25% | 10% | 1% |
|---|---|---|---|---|---|
| SVHN | FullyMap, $p = 53$ | FullyMap, $p = 51$ | FullyMap, $p = 49$ | FullyMap, $p = 80$ | FullyMap, $p = 80$ |
| CIFAR10 | IterMap-1, $p = 22$ | FreqMap-1, $p = 16$ | FreqMap-5, $p = 16$ | IterMap-5, $p = 16$ | IterMap-1, $p = 16$ |
| Flowers102 | FullyMap, $p = 16$ | FreqMap-1, $p = 16$ | FreqMap-1, $p = 16$ | IterMap-1, $p = 16$ | FullyMap, $p = 16$ |
| Food101 | FreqMap-1, $p = 16$ | IterMap-5, $p = 16$ | IterMap-1, $p = 16$ | IterMap-1, $p = 16$ | FullyMap, $p = 16$ |
| UCF101 | FullyMap, $p = 17$ | FullyMap, $p = 16$ | IterMap-1, $p = 16$ | FullyMap, $p = 16$ | FullyMap, $p = 16$ |
| OxfordIIITPet | FreqMap-10, $p = 16$ | FreqMap-5, $p = 16$ | FreqMap-5, $p = 16$ | FullyMap, $p = 16$ | FullyMap, $p = 16$ |
| CIFAR100 | FullyMap, $p = 30$ | FullyMap, $p = 26$ | IterMap-1, $p = 16$ | FreqMap-1, $p = 16$ | FullyMap, $p = 16$ |
| EuroSAT | FullyMap, $p = 16$ | FullyMap, $p = 16$ | FullyMap, $p = 16$ | FullyMap, $p = 16$ | FullyMap, $p = 16$ |
| DTD | FullyMap, $p = 18$ | IterMap-10, $p = 16$ | IterMap-5, $p = 16$ | FullyMap, $p = 16$ | FullyMap, $p = 16$ |
| ISIC | IterMap-1, $p = 16$ | FreqMap-1, $p = 16$ | FullyMap, $p = 80$ | FreqMap-5, $p = 48$ | FullyMap, $p = 80$ |
| FMoW | FullyMap, $p = 16$ | FullyMap, $p = 16$ | FullyMap, $p = 16$ | FreqMap-10, $p = 16$ | FullyMap, $p = 17$ |
| GTSRB | FullyMap, $p = 80$ | FullyMap, $p = 16$ | FullyMap, $p = 79$ | FullyMap, $p = 48$ | FullyMap, $p = 80$ |

## C.3 DOWNSTREAM DATASET ANALYSIS (ID/OOD *vs.* ACCURACY GAIN)

To further understand how the downstream dataset distribution influences the performance of visual prompting. We conduct experiments to observe the relation between accuracy gain and dataset characteristics. When using CLIP as the pre-trained model, we define *confidence score*, obtained by averaging the maximum softmax probability of predictions across the entire training set, as an indicator of dataset characteristics. For other vision pre-trained models, the FID score is used to measure the dissimilarity between the downstream dataset and the ImageNet-1K dataset, i.e. the degree of out-of-distribution (OOD). We present the Confidence/FID scores of each dataset in Figure C.2, providing insights into their OOD characteristics. Furthermore, Figure C.3 demonstrates the accuracy gain for each dataset when using ResNet18 as the pre-trained model. The experimental results show that when using AutoVP, datasets with a **higher degree of OOD tend to benefit from more accuracy gains**.

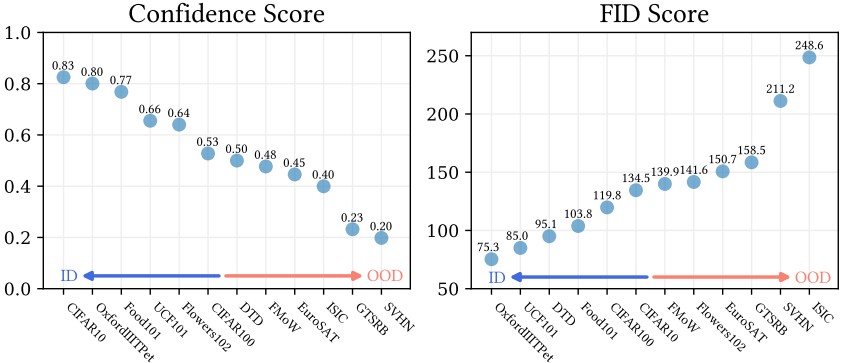

Figure C.2: **Confidence Score and FID Score.** We sort the datasets by the degree of out-of-distribution (OOD), where the one closer to the left indicates higher similarity (in-distribution, ID) to the training data of the pre-trained model, while the one closer to the right indicates greater dissimilarity (OOD).

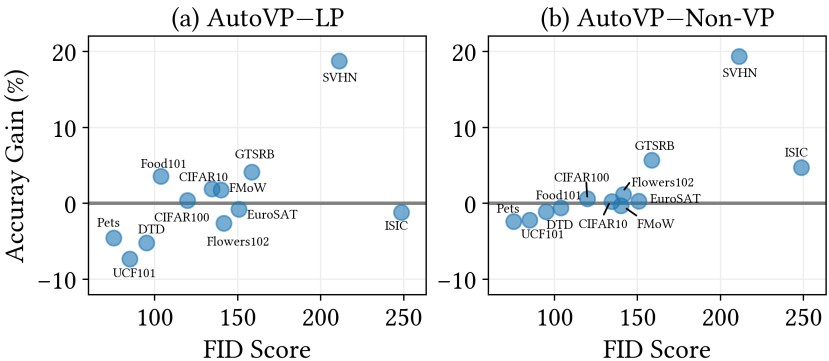

Figure C.3: **Accuracy Gains with Resnet18.** The gains are calculated by taking the difference between the performance of AutoVP and LP or Non-VP scenario.

Table 2 has encompassed the prevalent 12 datasets in VP research. Furthermore, to enable a more comprehensive comparison within a wider spectrum of VP research, we have included additional datasets—SUN397(Xiao et al., 2010), RESISC(Cheng et al., 2017), CLEVR(Johnson et al., 2017), and StanfordCar(Krause et al., 2013)—with their respective results available in Table C.4. Our findings consistently demonstrate AutoVP's superior performance compared to ILM and CLIP-VP across the most of datasets. Especially notable is the substantial 15% increase in accuracy observed in the out-of-distribution (OOD) dataset, CLEVR, when compared to linear probing (LP). However, the in-distribution (ID) datasets, SUN397 and StanfordCar, showcased lower performance than LP, aligning with the trend of accuracy gain illustrated in Figure 5 in Section 5.

Table C.4: **Performance on Additional Datasets.** The testing accuracy (%) for four additional datasets commonly found in VP research. The figure on the right side shows the accuracy gains.

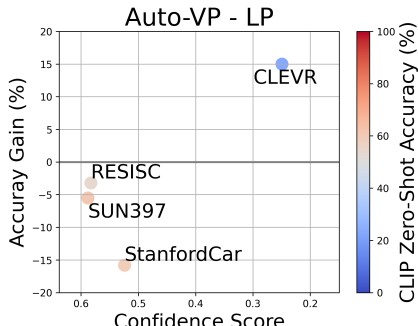

| Dataset | AutoVP Setting | AutoVP | LP | ILM | CLIP-VP |
|---------|---------------|--------|------|------|---------|
| SUN397 | FullyMap, $p = 16$ | **65.4** | 70.9 | 61.2 | 60.5 |
| StanfordCar | FullyMap, $p = 16$ | **61.8** | 77.6 | 57.6 | 56.2 |
| RESISC | FullyMap, $p = 17$ | **88.5** | 91.7 | 86.6 | 84.5 |
| CLEVR | FullyMap, $p = 16$ | 83.0 | 68.0 | **83.1** | 81.4 |

# D  ANALYSIS OF AUTOVP RESULTS

## D.1  PROMPTS IN FREQUENCY DOMAIN

We have analyzed the learned prompts using the best setting selected from AutoVP and have represented them in the frequency domain through Fast Fourier transformation (Brigham, 1988). In Figure D.1(a), the prompting result of SVHN dataset with CLIP is notably distinct and achieves the highest testing accuracy among all the pre-trained models. In the frequency analysis (Figure D.1(b)), the prompts are concentrated in the low-frequency domain (at the center of the plot), with CLIP displaying the most distinct structure and significantly larger magnitudes compared to the others. These results validate the efficient learning of prompts with CLIP, harnessing low-frequency features that generalize to the target domain.

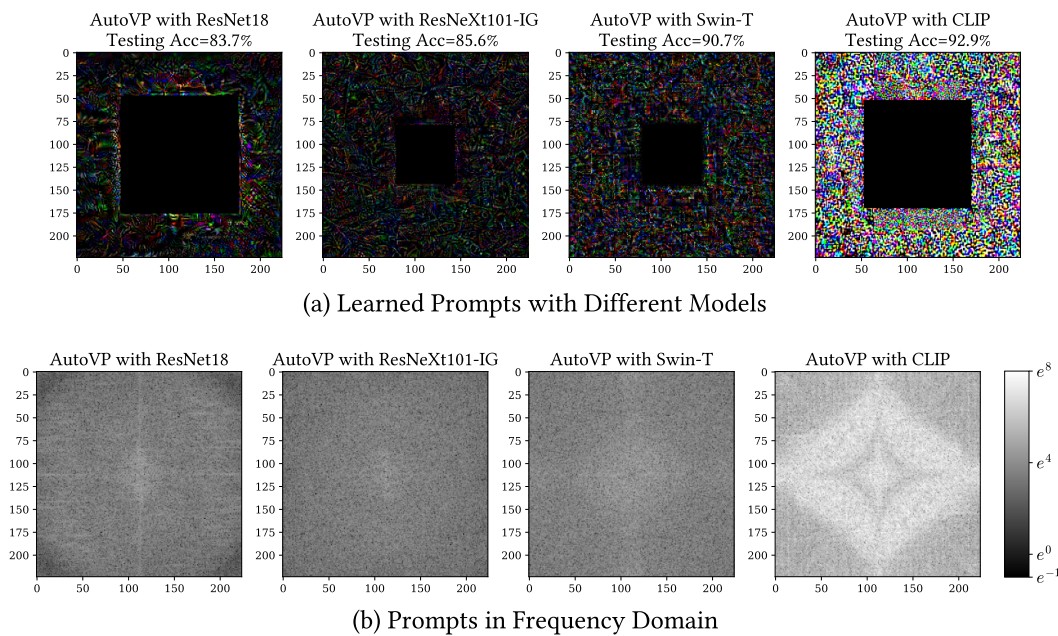

(a) Learned Prompts with Different Models

(b) Prompts in Frequency Domain

Figure D.1: **Prompting Analysis of SVHN Using Various Pre-trained Models.** (a) Frame-shape prompts learned with the best settings selected from AutoVP for a given pre-trained model. (b) Prompts in the frequency domain by Fast Fourier transformation.

## D.2 OUTPUT MAPPING ANALYSIS

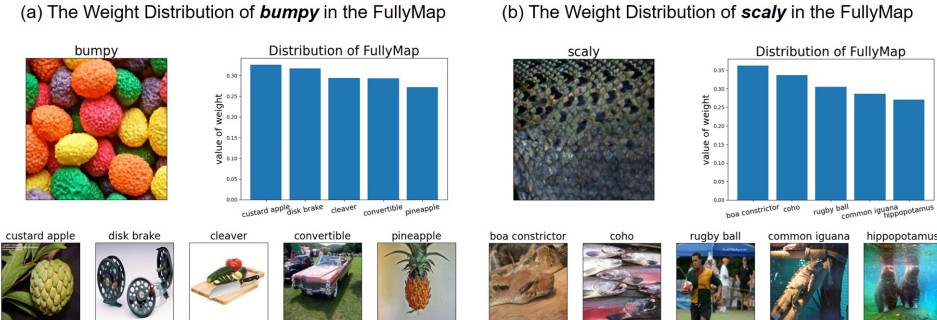

Figure D.2: **The Weight Distribution of FullyMap in DTD.** The top 5 source labels exhibiting the highest weight values within the FullyMap pertain to (a) the label **bumpy** and (b) the label **scaly**.

Figure D.2 illustrates that FullyMap can be interpreted as a weighted combination of multiple source labels, where some human-readable features may exhibit similarity. For instance, in Figure D.2(a), *bumpy* shows similarities with *custard apple*, *disk brake*, and *pineapple*, while in Figure D.2(b), *scaly* shares similar features with *boa constrictor*, *coho*, and *common iguana*.

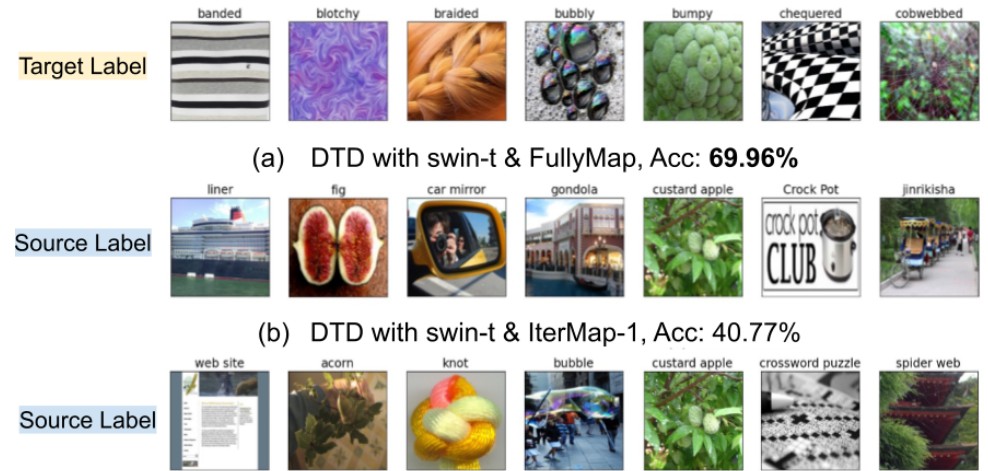

Figure D.3: **The Correspondence Between Output Mapping Labels in DTD.** Each column represents the mapping between the target label and its respective top-1 source label in FullyMap. (a) FullyMap (top-1 class having the largest weight) with Swin-T (second row). (b) IterMap-1 with Swin-T (third row).

Furthermore, when comparing FullyMap and IterMap, a significant accuracy gap is observed, with FullyMap achieving 69.96% and IterMap-1 only achieving 40.77%. However, in Figure D.3, IterMap has mapped to some classes that are indeed very close to the target. For instance, in Figure D.3(b), *braided* maps to *knot*, *bubbly* maps to *bubble*, and *cobwebbed* maps to *spider web*. This demonstrates that a mere combination of source labels is insufficient for achieving better performance; the weighting in the combination plays a crucial role, which is precisely what FullyMap accomplishes.

## D.3    THE PREFERENCES IN HYPER-PARAMETER TUNING SELECTION

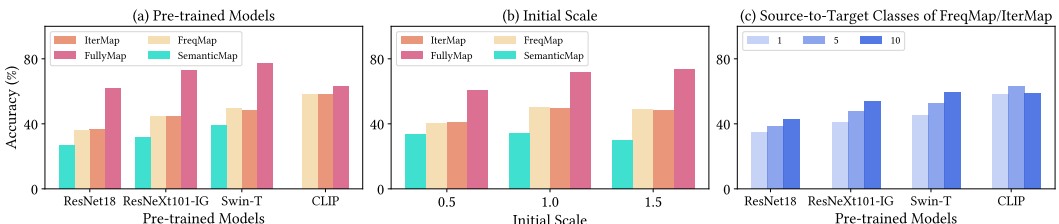

Figure D.4: **Average Tuning Results.** The three charts display the average few-epoch tuning accuracy for different selected conditions.

Figure D.4 illustrates the average tuning results for all datasets across different settings, including image scale, mapping methods, pre-trained models, and so on (see Figure A.2). In Figure D.4(a), among the vision models (ResNet18, ResNeXt-IG, Swin-T), the FullyMap demonstrates superior performance compared to the other methods. However, for CLIP, a vision-language model, the FullyMap only shows a slight advantage over the others. Furthermore, Figure D.4(b) indicates that larger initial scales, such as 1.0 or 1.5 (yielding square images with a width of $128 \times 1.0$ or $128 \times 1.5$), generally lead to better results when using FreqMap, IterMap, or FullyMap. Last, since both FreqMap and IterMap can configure the number ($m$) of source labels mapped to each target class, we found that increasing this count generally improves accuracy for the three vision models (see Figure D.4(c)). However, for CLIP, mapping five source labels appears to be the optimal choice based on the average tuning results.

## E    ABLATION STUDIES

### E.1    THE IMPACT OF TEXT ENCODER IN CLIP

Figure E.1 illustrates the impact on accuracy when incorporating or excluding the CLIP text encoder. On average, this configuration results in a significant decrease in accuracy of approximately 12%, highlighting the crucial role of the text encoder in VP with CLIP.

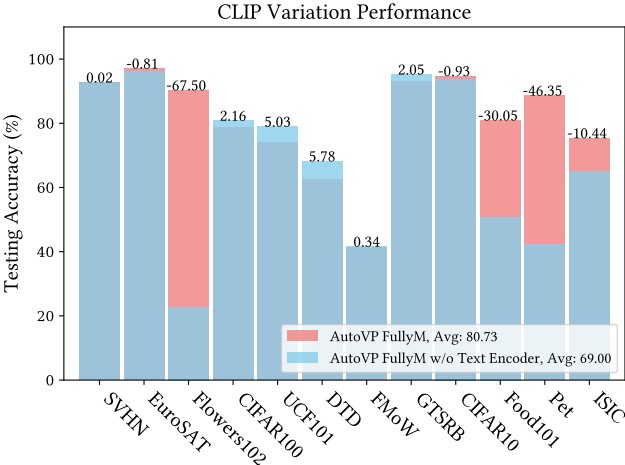

Figure E.1: **The Accuracy Difference of AutoVP Variations.** The chart shows the difference in accuracy between AutoVP variations with and without the CLIP text encoder. The numbers above the bars indicate the accuracy difference.

## E.2 Visual Prompting in Segmentation and Detection Tasks

Figure E.2: **LP and AutoVP in Segmentation and Detection Tasks.**

Although AutoVP primarily focuses on classification tasks, we aimed to delve deeper into various vision tasks by incorporating models designed for different purposes. For segmentation tasks, we employ DeepLabV3 (Chen et al., 2017) as the pre-trained backbone, and for detection tasks, we utilize OWL-ViT (Minderer et al., 2022). To evaluate the capability of both in-distribution (ID) and out-of-distribution (OOD) datasets, we chose ISIC as the OOD dataset for both tasks. For ID datasets, we use Pets in the segmentation task and VOC (Everingham et al., 2015) in the detection task.

Our VP segmentation framework, depicted in Figure E.2 (a), integrates a FullyMap after the pre-trained model to facilitate pixel-wise classification using a custom class number. In contrast, the linear probing approach modifies the final 2D convolutional layer, and the results are depicted in Table E.1.

Table E.1: **Performance on Segmentation Datasets.** The IoU and pixel accuracy (%) of linear probing (LP) and AutoVP.

| Dataset | LP | AutoVP |
|---------|----|--------|
| Pets | IoU : **0.83**, Pixel : **90.7%** | IoU : 0.77, Pixel : 86.9% |
| ISIC | IoU : 0.64, Pixel : 78.1% | IoU : **0.81**, Pixel : **89.5%** |

We evaluated segmentation performance using two metrics: IoU (Intersection over Union) score and pixel-wise classification accuracy. AutoVP exhibited superior performance on both metrics on the ISIC dataset. Additionally, segmentation examples highlighted that predictions align more accurately with the ground truth mask when the prompt space is larger (see Figure E.3). However, in the ID dataset (Pets), VP performance was inferior to LP. This aligns with our findings in the classification task, where OOD datasets derived greater benefits from visual prompts.

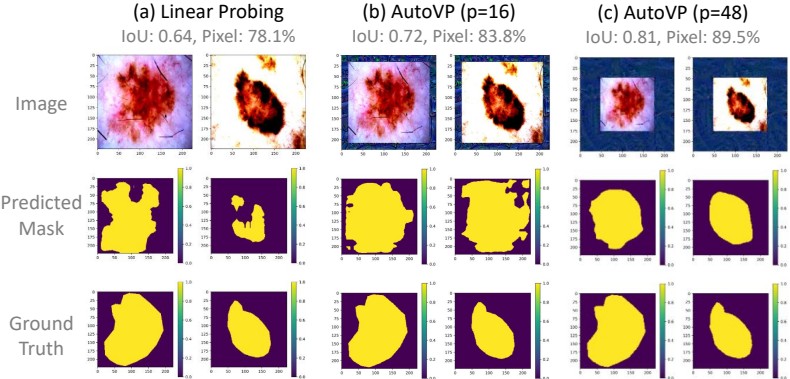

Figure E.3: **ISIC Segmentation.** Performance of LP and AutoVP with different prompt sizes ($p$). It's important to note that the performance calculation involves only the target image itself without the prompts, ensuring a fair comparison to LP. Specifically, the predicted mask is initially cropped to the region of the original target image and then resized to match the dimensions used in LP.

In the case of the detection task, as illustrated in Figure E.2 (b), the LP approach fine-tunes on the box predictor head, whereas the VP method incorporates trainable visual prompts. To achieve consistent box output, we have omitted the inclusion of output mapping. The results are shown in Table E.2. Similar to segmentation, the OOD dataset, ISIC, exhibits better performance than LP with a larger prompt size.

Table E.2: **Performance on Detection Datasets.** The IoU scores of linear probing (LP) and AutoVP.

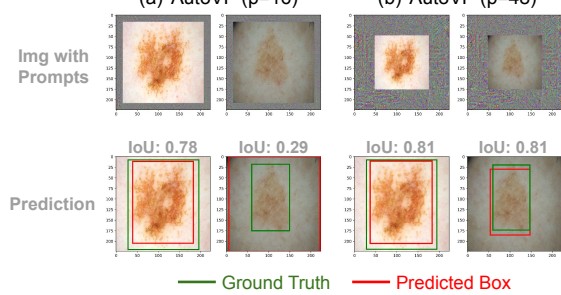

| Dataset | Zero-shot | Finetune box head | VP |
|---------|-----------|-------------------|-----|
| VOC | IoU: 0.75 | IoU: **0.76** | IoU: 0.73 |
| ISIC | IoU: 0.52 | IoU: 0.70 | IoU: **0.80** |

### E.3 EXPLORING ADDITIONAL TUNING AXES

In AutoVP, we set the learning rate (LR) for CLIP to be 40 and employed the SGD optimizer with a momentum of 0.9 and a weight decay (WD) of 0. In this section, we have undertaken supplementary tuning options within the AutoVP framework. Specifically, we introduce additional choices for the LR, selecting from 35, 40, and 45 (for CLIP), as well as WD, choosing from 0, $10^{-5}$, and $10^{-10}$. The outcomes presented in Table E.3 showcase a 0.6% enhancement in accuracy with the additional tuning options. It is important to note that the execution workload dramatically increases, as it will involve exploring 9 times more additional combinations in the tuning process.

Table E.3: **Hyper-Parameter Tuning for Learning Rate (LR) and Weight Decay (WD).** The table displays the optimal tuning configurations and corresponding testing accuracy for both scenarios with and without additional LR and WD tuning options for AutoVP on Flowers102. The pre-trained model used is CLIP. The highest accuracy is marked in **bold**.

| Setting | Tuning Selection | Testing Accuracy (%) |
|---|---|---|
| AutoVP w/o LR & WD tuning | FullyMap, $p = 16$
LR $= 40$
WD $= 0$ | 90.4 |
| AutoVP w/ LR & WD tuning | FullyMap, $p = 16$
LR $= 45$
WD $= 10^{-10}$ | **91.0** |

### E.4 IMPROVED ILM-VP WITH TUNING CONFIGURATION

In order to establish an equitable comparison with AutoVP, we undertake a comprehensive hyper-parameter tuning process for ILM-VP. We investigate tuning options that span 1, 2, 5, and 10 source classes for output mapping, as well as prompt sizes of 10, 20, 30, 40, and 50. For ILM-VP without tuning, default settings are maintained, with the mapping number set to 1 and the prompt size set to 30.

The tuning procedure is applied to several datasets, and the results are shown in Table E.4. Although hyper-parameter tuning in ILM-VP leads to an improvement in accuracy, it remains unable to surpass the performance exhibited by AutoVP.

Table E.4: **ILM-VP with Hyper-Parameter Tuning.** The chosen tuning configurations and the corresponding testing accuracy (%) in ILM-VP. The highest accuracy is marked in **bold**.

| Datasets | AutoVP | ILM-VP
w/ Tuning | ILM-VP
w/o Tuning |
|---|---|---|---|
| UCF101 | FullyMap, $p = 16$
Accuracy: **73.1**% | mapping number = 5, $p = 30$
Accuracy: 70.4% | Accuracy: 68.4% |
| EuroSAT | FullyMap, $p = 16$
Accuracy: **96.8**% | mapping number = 2, $p = 30$
Accuracy: 96.2% | Accuracy: 96.7% |
| OxfordIIITPet | FreqMap-10, $p = 16$
Accuracy: **88.2**% | mapping number = 5, $p = 30$
Accuracy: 86.7% | Accuracy: 85.1% |

### E.5 COMPARISON OF AUTOVP AND BLACKVIP

Table E.5: **AutoVP vs. BlackVIP.** The testing accuracy (%) of AutoVP and BlackVIP.

|  | Pets | Cars | Flowers | Food | SUN | DTD | SVHN | EuroSAT | RESISC | CLEVR | UCF | **Avg.** |
|---|---|---|---|---|---|---|---|---|---|---|---|---|
| **AutoVP** | 88.2 | 61.8 | 90.4 | 82.3 | 65.4 | 62.5 | 92.9 | 96.8 | 88.5 | 82.8 | 73.1 | **80.4** |
| **BlackVIP** | 89.7 | 65.6 | 70.6 | 86.6 | 64.7 | 45.2 | 44.3 | 73.1 | 64.5 | 36.8 | 69.1 | **64.6** |

Some visual prompting research has delved into a black-box setting (Tsai et al., 2020; Oh et al., 2023), where the internal architecture of the pre-trained model remains unattainable during the training process. For instance, BlackVIP (Oh et al., 2023) employs an input-dependent prompt designer to generate visual prompts. These prompts are then fed into a black-box model, and gradient approximation strategies are utilized to update the prompt designer. In Table E.5, we present a comparison between AutoVP and BlackVIP. Since BlackVIP uses CLIP as the pre-trained backbone, we report our results using the same pre-trained model. AutoVP demonstrates a 16% performance increase on average compared to BlackVIP. This notable difference might be attributed to the variance in update strategies: BlackVIP utilizes SPSA-GC for black-box models, while AutoVP relies on classic gradient descent. However, due to the differing objectives of these two studies, direct comparisons may introduce certain unfairness.

## F PERFORMANCE AND RESOURCE UTILIZATION

### F.1 COMPARISON OF AUTOVP, LINEAR PROBING, AND FULL FINE-TUNING

When comparing the performance of AutoVP to that of linear probing (LP) and full fine-tuning (FF), it's crucial to acknowledge the significant discrepancy in terms of trainable parameter size. As FF involves a parameter size that is roughly 100 to 1000 times larger (as shown in Table F.4). Nevertheless, it's worth noting that the differences in performance between VP and FF are relatively minor in certain datasets. For instance, in the case of EuroSAT, both AutoVP and FF achieve a 96% accuracy (as demonstrated in Table F.1). This observation suggests that VP retains its advantages even when faced with such substantial variations in parameter size.

Table F.1: **Comparison of AutoVP, Linear Probing (LP), and Full Fine-Tuning (FF).** With the EuroSAT dataset and using CLIP as the pre-trained model, the table displays the testing accuracy, execution time, and trainable parameter size associated with each respective method.

| Experimental Info. | AutoVP | LP | FF |
|---|---|---|---|
| Accuracy (%) | 96.84 | 94.62 | 96.78 |
| Execution Time (second) | 2448 | 2370 | 3081 |
| Trainable Parameter Size (Million) | 0.15 | 0.005 | 151.28 |

## F.2 COMPUTING RESOURCES

In this section, we provide the execution time (measured on NVIDIA GeForce RTX 3090) and the comparison of trainable parameters of AutoVP. For ease of comparison, we use the Flower102 dataset for illustration. In Table F.2, we provide the end-to-end execution time (hyper-parameter tuning + 200-epoch training) of AutoVP. When we measure only the 200-epoch training, AutoVP demonstrates its competitiveness in terms of similar or even lower training time (see Table F.3), compared to the state-of-the-art VP baselines (ILM-VP and CLIP-VP The comparison of trainable parameters can also be found in Table F.4.

Table F.2: **Flowers102 End-to-End Execution Time.** For the Flowers102 dataset, the pre-trained model selected by AutoVP is Swin-T, the output mapping is FullyMap, and the prompt size is 16.

|  | Hyper-Parameter Tuning | 200-Epoch Training | Total Execution Time |
|---|---|---|---|
| Time | 146 mins | 31 mins | 177 mins |

Table F.3: **Execution Time Comparison.** The 200-epoch training time (in minutes) on the Flowers102 dataset varies depending on the chosen pre-trained model: Swin-T or CLIP. In both cases, AutoVP utilizes the FullyMap as the output mapping method with the prompt size 16.

| Pre-trained model | AutoVP | ILM-VP | CLIP-VP | Linear Probing |
|---|---|---|---|---|
| Swin-T | 31 | 43 | — | 28 |
| CLIP | 39 | 76 | 38 | 40 |

Table F.4: **Trainable Parameter Size.** The average trainable parameter sizes (million) are calculated across the 12 datasets for different pre-trained models, mapping methods, and baselines.

|  | AutoVP | | | | Linear Probing | Full Finetune |
|---|---|---|---|---|---|---|
|  | SemanticMap | FreqMap | IterMap | FullyMap |  |  |
| ResNet18 | 0.15 | 0.15 | 0.15 | 0.20 | 0.03 | 11.20 |
| ResNeXt101-IG | 0.15 | 0.15 | 0.15 | 0.20 | 0.11 | 86.85 |
| Swin-T | 0.15 | 0.15 | 0.15 | 0.20 | 0.04 | 27.56 |
| CLIP | 0.15 | 0.15 | 0.15 | 0.49 | 0.03 | 151.23 |

