# OpenReview forum: "AutoVP: An Automated Visual Prompting Framework and Benchmark"
_ICLR.cc/2024/Conference — ICLR 2024 poster_

### Official Review · Reviewer_aYHC · 2023-10-22

**Soundness:** 3 good
**Presentation:** 4 excellent
**Contribution:** 2 fair
**Rating:** 6
**Confidence:** 5

**Summary:**

This paper mainly focuses on visual prompting method. The authors propose a new framework which covers a wide range of design space for visual prompting including input scaling, pretrained model, output mapping. The authors introduce several difference techniques for these dimension which are shown to be effective in the experiments.

**Strengths:**

1. The proposed method is simple yet effective.
2. The writing is easy to understand.

**Weaknesses:**

1. The input scaling technique was discussed in EVP [1]. The authors may have some more discussion about the relationship and difference with this work.
2. The different choices for pretrained model are almost the same as the original ones in VP. It would be better if the author can mention this in the paper.
3. I doubt if the proposed FullyMap can be viewed as an output mapping method rather than a variant of linear probing. By using this method, the distribution of trainable parameters is totally different from the original VP-style methods like VP, EVP and BlackVIP [2] whose trainable parameters only concentrate on modifying the input space without the output space. It is for sure that such a method can bring significant improvement in the experiments. Even if you do not use the visual prompt but simply finetune the model with the additional last fc layer, I assume the model can have similar performance to the linear probed one.
4. I recommend the authors adopt other datasets used in previous VP papers like SUN, CLEVR and RESISC, and compare the proposed method with other baselines like BlackVIP.


[1] Wu J, Li X, Wei C, et al. Unleashing the power of visual prompting at the pixel level[J]. arXiv preprint arXiv:2212.10556, 2022.
[2] Oh C, Hwang H, Lee H, et al. BlackVIP: Black-Box Visual Prompting for Robust Transfer Learning[C]//Proceedings of the IEEE/CVF Conference on Computer Vision and Pattern Recognition. 2023: 24224-24235.

**Questions:**

Please refer to the weaknesses.

---

> ### Author Response · Authors · 2023-11-20
> **Response to Reviewer aYHC (1 of 4)**
>
> Dear Reviewer aYHC,
>
> Thank you for your time and effort in reviewing our paper. We appreciate that you enjoyed reading our paper, and recognized our proposed framework is simple yet effective. To your questions, we address our response in the following.
>
> **Q1. More Discussion on EVP**
>
> Thank you for your thoughtful reminder! EVP is indeed a very inspirational paper that motivates our work. While we did cite EVP, our discussion of their findings on input scaling was omitted. We referenced their findings in Section 3. It's important to highlight that the input scaling in AutoVP extends beyond determining the size of prompt frames and resizing images accordingly. We integrate a **dynamic optimization method** during the training process, elaborated in Section 3: Input Scaling. This represents the most significant departure from EVP, which merely shrinks the input image and fixes the prompt size. Our design optimizes both prompts and prompt sizes, and adjusts the input image accordingly, maximizing the effectiveness of prompts.

---

> ### Author Response · Authors · 2023-11-20
> **Response to Reviewer aYHC (2 of 4)**
>
> **Q2. The Choices for Pre-trained Model**
>
> Within the AutoVP framework, we have opted to utilize state-of-the-art classification models, comprising residual network-based, vision transformer-based, and multi-modal models (such as CLIP), within our pre-trained classifier set. The decision to select similar pre-trained backbones aims to ensure equitable comparisons with previous studies, as highlighted in the fixed model comparison in Tables 2 and 8 of the paper. However, we have also integrated new pre-trained models, such as **Swin-T**. Notably, in Table 9 -- AutoVP with source model selection, Swin-T emerges as the most frequently chosen model, illustrating its significant role that contributes to the overall performance of AutoVP.

---

> ### Author Response · Authors · 2023-11-20
> **Response to Reviewer aYHC (3 of 4)**
>
> **Q3. FullyMap and LP**
>
> While LP and FullyMap share similarities due to both incorporating a linear head, we conducted a comparison between LP and pure FullyMap (without visual prompts). The results shown in the **table below**, demonstrate that **FullyMap without VP exhibits inferior performance compared to LP** on most datasets. This discrepancy may arise from the inability of the pre-trained model's learned features in the linear head to be interpreted for downstream classes, highlighting why FullyMap is not equivalent to LP.
>
> For a more detailed exploration of FullyMap, **Appendix B.10** Figure 15 illustrates that FullyMap can be interpreted as a weighted combination of multiple source labels, where some human-readable features may exhibit similarity. For instance, in Figure 15(a), 'bumpy' shows similarities with 'custard apple,' 'disk brake,' and 'pineapple,' while in Figure 15(b), 'scaly' shares similar features with 'boa constrictor,' 'coho,' and 'common iguana.'
>
> While Figure 16(b) portrays IterMap as successfully mapping to classes more closely aligned with the target classes than FullyMap (Figure 16(a)), a significant accuracy gap is evident. In Figure 16, FullyMap achieves 69.96% accuracy, whereas IterMap-1 only reaches 40.77%. This highlights that merely combining source labels is insufficient for superior performance; **the weighting within the combination plays a crucial role, precisely what FullyMap achieves**.
>
> |   Dataset  | Linear Probing | Non-VP + FullyMap | FullyMap - LP |
> |:----------:|:--------------:|:-----------------:|:-------------:|
> |    SVHN    |      65.4      |        34.2       |     -31.2     |
> | Flowers102 |      96.9      |        89.1       |      -7.8     |
> |   UCF101   |      83.3      |        80.2       |      -3.1     |
> |    Pets    |      89.2      |        91.3       |      +2.1     |
> |  CIFAR100  |      80.0      |        74.5       |      -5.5     |
> |   EuroSAT  |      95.3      |        73.2       |     -21.1     |
> |     DTD    |      74.6      |        67.7       |      -6.9     |
> |    ISIC    |      71.9      |        42.6       |     -29.3     |
> |    FMoW    |      36.3      |        34.9       |      -1.4     |
> |    GTSRB   |      85.8      |        66.0       |     -19.8     |

---

> ### Author Response · Authors · 2023-11-20
> **Response to Reviewer aYHC (4 of 4)**
>
> **Q4-1. Other Vision Datasets**
>
> We appreciate the reviewer's mention of these prominent datasets. We incorporated additional datasets—**SUN397, RESISC, CLEVR, and StanfordCar** —and their results are available in **Appendix B.4 Table 5**.
>
> Our findings demonstrate that AutoVP generally outperforms ILM-VP and CLIP-VP across most datasets. When compared to linear probing (LP), the out-of-distribution (OOD) dataset, CLEVR, demonstrated a significant 15% increase in accuracy. However, the in-distribution (ID) datasets, SUN397 and StanfordCar, show inferior performance than LP. This aligns with our findings discussed in Section 5: Performance Evaluation on ID/OOD.
>
> **Q4-2. Comparison with BlackVIP**
>
> We thank the reviewer for pointing out BlackVIP, one of the pioneering works in VP. However, we did not compare it with AutoVP because BlackVIP operates in a black-box setting, treating the pre-trained backbone as a black box. In contrast, AutoVP views pre-trained backbones as white-box models. Notably, AutoVP focuses on the concept of 'reprogramming' an online API using publicly released model architecture and weights, even though direct access to the model itself is not possible.
>
> Despite this distinction, we present a comparison across 11 datasets with BlackVIP, and the results are summarized in the table below. Since BlackVIP selects CLIP as the pre-trained backbone, we report our results under the same pre-trained model. Our results demonstrate a **16% outperformance compared to BlackVIP**. This considerable gap might be attributed to the difference in *update strategies*: BlackVIP employs SPSA-GC for black-box models, whereas AutoVP utilizes classic gradient descent. Given the differing objectives of these two studies, direct comparisons may present certain unfairness. This discussion has been added in **Appendix B.14**.
>
> |          | Pets | Cars  | Flowers | Food | SUN   | DTD   | SVHN | EuroSAT | RESISC | CLEVR | UCF  | Avg.  |
> |----------|------|-------|---------|------|-------|-------|------|---------|--------|-------|------|-------|
> | AutoVP   | 88.2 | 61.82 | 90.4    | 82.3 | 65.42 | 62.5  | 92.9 | 96.8    | 88.46  | 82.76 | 73.1 | 80.42 |
> | BlackVIP | 89.7 |  65.6 | 70.6    | 86.6 | 64.7  | 45.2  | 44.3 | 73.1    | 64.5   | 36.8  | 69.1 | 64.56 |

---

> ### Comment · Reviewer_aYHC · 2023-11-22
>
> Thank the authors for the feedback. Most of my questions have been solved except for Q3. It is still not clear how such a technique can be viewed as an improved version of VP since they change totally different layers in a model, one for input and one for the prediction.

---

> ### Author Response · Authors · 2023-11-22
> **Thank you for your positive feedback!**
>
> Dear Reviewer aYHC,
>
> We are delighted to learn that "Most of my questions have been solved except for Q3". We understand your concern when we cast the fully map as a generalized/improved version of label mapping (i.e., soft label mapping with learnable weights). We will modify the wording in the future version to clarify this point.
>
> To address your remaining concern, we would like to bring to your attention that AutoVP is indeed an improved version of existing VP methods, when we constrain the label mapping to be the same as baseline methods, such as IterMap and FreqMap. For example, in the table below (extracted from Table 2), when setting the same label mapping as IterMap-1, one can see that AutoVP improves the accuracy of ILM-VP. The reason is that the feature of input scaling optimization enabled by AutoVP contributes to improved accuracy. Similarly, when including FreqMap-1 (an existing label mapping method) in AutoVP, we also observe an obvious accuracy boost over ILM-VP, such as the Food101 dataset in the table below.
>
> We hope these case studies can convince the reviewer that AutoVP is indeed an improved version of VP. We are at the reviewer's disposal to answer any remaining concerns the reviewer may have.
>
> |    Dataset    |   AutoVP Setting   |   AutoVP    | ILM-VP  |
> |:-------------:|:------------------:|:-----------:|:-------:|
> |    CIFAR10    |  IterMap-1, p = 23 |  95.2 ± 0.0 |  94.4   |
> |      ISIC     |  IterMap-1, p = 16 | 74.0 ± 1.0  |   73.3  |
> |    Food101    | FreqMap-1, p = 16  | 82.3 ± 0.1  |   79.1  |

---

> ### Author Response · Authors · 2023-11-23
> **Follow up to the Reviewer**
>
> Dear Reviewer aYHC,
>
> As the discussion period is closing soon, we'd like to follow up and see if the Reviewer has had a chance to consider our response. The additional experimental results on the detection task have been provided in our [Update Response](https://openreview.net/forum?id=wR9qVlPh0P&noteId=vayyWGrn3M). We hope our responses and new results are helpful for finalizing the review rating.
>
> Yours Sincerely,
>
> Authors

---

> > ### Comment · Reviewer_aYHC · 2023-11-23
> >
> > Thank the authors for providing comprehensive feedback. Personally I support the acceptance of this paper, since my questions have been solved.

---

> > > ### Author Response · Authors · 2023-11-23
> > >
> > > We thank the reviewer for the prompt response! We are glad our response could address your concerns. Thank you for the endorsement of recommending acceptance.

---

### Official Review · Reviewer_dWYk · 2023-11-01

**Soundness:** 3 good
**Presentation:** 3 good
**Contribution:** 2 fair
**Rating:** 3
**Confidence:** 3

**Summary:**

The paper introduces AutoVP, an end-to-end framework that automatically selects optimal visual prompt (VP) configurations for vision downstream datasets. Tested on 12 datasets, AutoVP demonstrates improvements over the traditional linear probing approach, and on OOD test. The performance enhancements are in line with expectations. The significance of AutoVP lies in its automation of the VP configuration process, offering an extensive study for prompt engineering with gains.

**Strengths:**

1. An extensive study for visual prompting on vision model such as ResNeXt, ViT, and CLIP model.

2. AutoVP, by applying a series of established approach, from input scaling, to output label engineering, enables huge gain on the results.

**Weaknesses:**

1. The paper is evaluated on 12 visual recognition tasks, what about other tasks, given that this is a benchmark paper. Say Object Detection, Depth, Segmetnation.

2. Reviewer appreciate this systematic study in applying all methods of VP and improve results. However, those results are expected.  Learn a bit of new knowledge after reading this, the reviewer would expect in general more surprising finding or impressive knowledge.

**Questions:**

No

---

> ### Author Response · Authors · 2023-11-20
> **Response to Reviewer dWYk (1 of 2)**
>
> Dear Reviewer dWYk,
>
> Thank you for your time and effort in reviewing our paper. To your questions, we address our response in the following.
>
> **Q1. Variety of Vision Tasks**
>
> We appreciate the reviewer's interest in exploring AutoVP’s performance on other vision tasks. We acknowledge the need for a closer examination of various vision tasks. To address this, we integrated AutoVP into a **segmentation task** and compared its performance with linear probing on both ID and OOD datasets. The results are provided in **Appendix B.6** and table below.
>
> In our VP segmentation framework illustrated in Figure 12, a FullyMap is incorporated after the pre-trained model to enable pixel-wise classification with a custom class number. In contrast, the linear probing approach modifies the last 2D convolutional layer. Table 6 displays the outcomes.
>
> We evaluated segmentation performance using two metrics: IoU (Intersection over Union) score and pixel-wise classification accuracy. **AutoVP exhibited superior performance on both metrics in the ISIC dataset**. Additionally, segmentation examples highlighted that predictions align more accurately with the ground truth mask when the prompt space is larger (see Figure 13). However, in the ID dataset (Pets), VP performance was inferior to LP. This aligns with our findings in the classification task, where OOD datasets derived greater benefits from visual prompts.
>
> | Dataset |             LP            |           AutoVP          |
> |:-------:|:-------------------------:|:-------------------------:|
> |   Pets  | IoU : 0.83, Pixel : 90.7% | IoU : 0.77, Pixel : 86.9% |
> |   ISIC  | IoU : 0.64, Pixel : 78.1% | IoU : 0.81, Pixel : 89.5% |
>
> As for **detection tasks**, we are still working on the experiments. We aim to provide those results before the end of the author-reviewer rebuttal deadline.

---

> ### Author Response · Authors · 2023-11-20
> **Response to Reviewer dWYk (2 of 2)**
>
> **Q2. AutoVP Novelty and Insights**
>
> We understand that as a unified automated visual prompting (VP) framework, the reviewer may feel that it lacks originality because it contains many visual prompting methods as special cases. However, we would like to respectfully point out several new components and designs in our AutoVP that the reviewer may overlook, as well as highlight our major contributions to VP.
>
> **[Our novel designs for VP]** AutoVP introduces novel components that have not been studied in prior arts in VP, including the **automated input scaling**, as well as **weight initialization** when using FullyMap. These additions contribute to the uniqueness of our framework, positioning AutoVP not just as a mere automated tuning tool, but as an advanced and comprehensive framework. It amalgamates a multitude of techniques to discern optimal VP configurations for datasets that exhibit diverse characteristics. More importantly, we also show that these two components are very critical to improving VP performance. For example, the optimal configurations in Table 2 suggest that different datasets prefer distinct image scales, and weight initialization with FullyMap is one of the most frequently selected output mapping methods. In doing so, our work transcends the confines of being solely a tuning tool and stands as a powerful toolset for effectively developing and deploying VP across a range of scenarios.
>
> **[Our contribution in demonstrating the advantage of VP]** It's worth noting that one of our pivotal achievements is the notable enhancement in VP performance beyond the established linear probing baseline, which had not been demonstrated in prior arts  (ILM-VP [5] and CLIP-VP [2]). In particular, our data scalability analysis in Sec. 4.1 shows that AutoVP substantially outperforms LP in the few-shot settings (10% and 1% data usage). We believe this is a significant and exciting finding, because efficient few-shot learning is exactly the motivation for prompting.
>
> **[Our contribution in providing new insights for VP]** Another noteworthy result is **our comprehensive analysis of VP on the OOD/ID dataset**, where we show that using AutoVP can significantly improve accuracy (as showcased in Figure 5(b)). This outcome highlights the capacity of AutoVP to operate with a wider prompt space when dealing with OOD datasets like SVHN and GTSRB, thereby leading to substantial accuracy gains. This aspect underscores VP's inherent adaptability to different dataset characteristics and has not been systematically studied in the existing literature.
>
> In response to this comment, we highlighted the novelty of AutoVP in the revision within the following sections. In the introduction, we have emphasized the **novel components (automated input scaling and weight initialization)** in our main contributions to provide readers with a clear impression at the outset. Additionally, we delve into a detailed discussion of the contributions made by these new modules in Section 5. Furthermore, we've incorporated the reviewer's suggestions to integrate a **segmentation task** into our revised version, available in Appendix B.6.

---

> ### Author Response · Authors · 2023-11-23
> **Follow up to the Reviewer**
>
> Dear Reviewer dWYk,
>
> As the discussion period is closing soon, we'd like to follow up and see if the Reviewer has had a chance to consider our response. The additional experimental results on the detection task have been provided in our [Update Response](https://openreview.net/forum?id=wR9qVlPh0P&noteId=vayyWGrn3M). We hope our responses and new results are helpful for finalizing the review rating.
>
> Yours Sincerely,
>
> Authors

---

### Official Review · Reviewer_PZJW · 2023-11-04

**Soundness:** 3 good
**Presentation:** 3 good
**Contribution:** 2 fair
**Rating:** 5
**Confidence:** 4

**Summary:**

This paper introduces AutoVP, a comprehensive framework for automating VP design decisions. This framework covers several aspects, including optimizing the prompts, selecting suitable pre-trained models, and determining the best output mapping strategies.
The AutoVP along with a set of 12 image-classification tasks serves as a benchmark for evaluating VP performance.

AutoVP is shown to outperform existing VP methods and achieves a considerable performance boost when compared to the linear-probing baseline. Above all, AutoVP offers a hyperparameter tuning tool for VP and a benchmark to further its advancements.

**Strengths:**

1. This paper presents its findings with clear figures and detailed statistical reports, making it easier for readers to grasp the results.
2. This paper does not just present a tool but embarks on a detailed exploration of optimal configurations under various conditions, aiming at proving how different settings affect performance. It also examines the impact of domain similarity on VP performance.

**Weaknesses:**

1. While VP can potentially be used for a variety of vision tasks, the paper seems to focus primarily on image classification tasks, which may limit its applicability to broader vision problems. Are there any additional results on dense discriminant tasks?
2. When utilizing CLIP as the pre-trained classifier within the framework, which visual backbone is employed, ViT or ResNet?
3. About the proposed VP benchmark, why do the authors exclude some widely recognized 2D datasets, such as Caltech101, OxfordPets, StanfordCars, SUN397, EuroSAT, and FGVCAircraft, which are all common-used for 2D image classification task evaluation? What are the criteria for dataset selection?
4. Table 2 reveals that AutoVP underperforms on **6** datasets out of **12** in the benchmark compared to Linear Probing (LP), e.g., AutoVP is **-6.5%, -10.2%, -12.1%** lower than LP on Flowers102, UCF101, and DTD respectively. The reported average accuracy improvement appears to be significantly influenced by the results from the SVHN dataset, which is **+27.5%**. Could the authors provide additional insights into this discrepancy? Furthermore, similar patterns observed in Tables 6 and 7 suggest that these results may not be consistently solid across varied datasets.
5. The paper mentions ***std*** for AutoVP, indicating some randomness. It's important to think about how this inconsistency could affect the reliability of AutoVP, especially compared to the more stable method of linear probing. While Table 5 indicates a shorter execution time per run for AutoVP, one might infer that achieving the reported performance could necessitate multiple runs, thereby affecting the efficiency.  This inconsistency and the potential extra time needed make it less practical in certain situations.

**Questions:**

Please see the weaknesses mentioned above.

---

> ### Author Response · Authors · 2023-11-20
> **Response to Reviewer PZJW (1 of 5)**
>
> Dear Reviewer PZJW,
>
> Thank you for your time and effort in reviewing our paper. We appreciate that you enjoyed reading our paper. To your questions, we address our response in the following.
>
> **Q1. Variety of Vision Task**
>
> We focused on image classification as our work is mainly based on previous VP works that also cope with classification tasks. However, we acknowledge the need for a closer examination of various vision tasks. To address this, we integrated AutoVP into a **segmentation task** and compared its performance with linear probing on both ID and OOD datasets. The results are provided in **Appendix B.6** and table below.
>
> In our VP segmentation framework illustrated in Figure 12, a FullyMap is incorporated after the pre-trained model to enable pixel-wise classification with a custom class number. In contrast, the linear probing approach modifies the last 2D convolutional layer. Table 6 displays the outcomes.
>
> We evaluated segmentation performance using two metrics: IoU (Intersection over Union) score and pixel-wise classification accuracy. **AutoVP exhibited superior performance on both metrics in the ISIC dataset**. Additionally, segmentation examples highlighted that predictions align more accurately with the ground truth mask when the prompt space is larger (see Figure 13). However, in the ID dataset (Pets), VP performance was inferior to LP. This aligns with our findings in the classification task, where OOD datasets derived greater benefits from visual prompts.
>
> | Dataset |             LP            |           AutoVP          |
> |:-------:|:-------------------------:|:-------------------------:|
> |   Pets  | IoU : 0.83, Pixel : 90.7% | IoU : 0.77, Pixel : 86.9% |
> |   ISIC  | IoU : 0.64, Pixel : 78.1% | IoU : 0.81, Pixel : 89.5% |
>
>
> As for **detection tasks**, we are still working on the experiments. We aim to provide those results before the end of the author-reviewer rebuttal deadline.

---

> ### Author Response · Authors · 2023-11-20
> **Response to Reviewer PZJW (2 of 5)**
>
> **Q2. CLIP Backbone**
>
> We utilized CLIP with the **ViT-B/32** vision encoder backbone. We apologize for omitting this information; we have now included it in Section 3: Pre-trained Classifier.

---

> ### Author Response · Authors · 2023-11-20
> **Response to Reviewer PZJW (3 of 5)**
>
> **Q3. Other Vision Datasets**
>
> There are no specific criteria for selecting datasets, and the datasets were chosen mainly from state-of-the-art VP baselines [1]. We appreciate the reviewer's mention of prominent datasets. Among the mentioned datasets, such as OxfordPets and EuroSAT, these were already included in our chosen sets. Moreover, we incorporated additional datasets—**StanfordCars, SUN397, RESISC, and CLEVR**—and their results are available in **Appendix B.4 Table 5**.
>
> Our findings demonstrate that AutoVP generally outperforms ILM-VP and CLIP-VP across most datasets. When compared to linear probing (LP), the out-of-distribution (OOD) dataset, CLEVR, demonstrated a significant 15% increase in accuracy. However, the in-distribution (ID) datasets, SUN397 and StanfordCar, show inferior performance than LP. This aligns with the findings discussed in Section 5: Performance Evaluation on ID/OOD.
>
> [1] Chen et al. Understanding and Improving Visual Prompting: A Label-Mapping Perspective (CVPR 2023)

---

> ### Author Response · Authors · 2023-11-20
> **Response to Reviewer PZJW (4 of 5)**
>
> **Q4. Accuracy Improvement across Datasets**
>
> AutoVP has exhibited noteworthy accuracy improvements on out-of-distribution (OOD) datasets such as SVHN. However, it displayed comparatively lower performance on specific in-distribution (ID) datasets. The considerable variation in performance across datasets is rooted in the extent of dissimilarity between the source and target datasets. This aspect has been discussed in Section 5: Performance Evaluation on ID/OOD.

---

> ### Author Response · Authors · 2023-11-20
> **Response to Reviewer PZJW (5 of 5)**
>
> **Q5. Randomness of AutoVP and LP**
>
> We thank the reviewer for pointing out this concern. However, the fact that we do not report the standard deviations of other baselines does not mean that they are more stable than AutoVP, but that most results are adapted from original papers, where their authors also do not report them in the paper. We apologize for causing the misunderstanding to the reviewer. To address this, we have provided the standard deviation of ISIC and EuroSAT for both AutoVP and linear probing in the table below. The results show that **AutoVP and LP exhibit similar standard deviation**, suggesting the stability of AutoVP.
>
> |         | AutoVP      | LP          |
> |---------|-------------|-------------|
> |    ISIC | 74.0 ± 1.0  | 68.50 ± 1.4 |
> | EuroSAT | 96.8 ± 0.2  | 94.70 ± 0.1 |

---

> ### Author Response · Authors · 2023-11-23
> **Follow up to the Reviewer**
>
> Dear Reviewer PZJW,
>
> As the discussion period is closing soon, we'd like to follow up and see if the Reviewer has had a chance to consider our response. The additional experimental results on the detection task have been provided in our [Update Response](https://openreview.net/forum?id=wR9qVlPh0P&noteId=vayyWGrn3M). We hope our responses and new results are helpful for finalizing the review rating.
>
> Yours Sincerely,
>
> Authors

---

### Official Review · Reviewer_nUAn · 2023-11-10

**Soundness:** 3 good
**Presentation:** 3 good
**Contribution:** 2 fair
**Rating:** 8
**Confidence:** 4

**Summary:**

This paper proposes AutoVP, an end-to-end expandable framework for automating VP design choices along with 12 downstream image classification tasks that can serve as a holistic VP-performance benchmark.

**Strengths:**

**Clarity and Logic**: The paper is well-structured and presents complex ideas clearly, making it understandable for readers.

**Useful Framework**: AutoVP is introduced as a versatile toolbox that simplifies the development of visual prompts, offering a modular design and comprehensive functionalities.

**Improved Performance**: The models tuned with AutoVP demonstrate a significant performance improvement over previous baselines across various image classification tasks.

**Weaknesses:**

**Limited Novelty**: The framework largely combines existing methods, which might suggest a wrap-up of previous work rather than introducing new concepts, limiting the perceived novelty of the research.

**Potential Overfitting**: AutoVP uses different settings for different datasets, raising the question of whether these are overfitted to specific tasks and what the implications are for a robust, universal setting.

**Insufficient Analysis of Mapping Methods**: There is a lack of detailed comparison and analysis of the mapping methods used in visual prompting, which is necessary to understand their impact and provide more comprehensive insights.

**Questions:**

I would suggest expanding the testing of visual prompting from image classification to other tasks like detection and segmentation. This would help ensure that the AutoVP framework is versatile and not just fine-tuned for specific tasks. Aim to create a benchmark that evaluates how well visual prompting works generally, across various types of visual tasks.

---

> ### Author Response · Authors · 2023-11-20
> **Response to Reviewer nUAn (1 of 4)**
>
> Dear Reviewer nUAn,
>
> Thank you for your encouraging review, and for recognizing our paper is well-written, the proposed toolbox is useful and has demonstrated its state-of-the-art efficacy across various image classification tasks. We are thrilled that you enjoyed our paper. To your questions, we address our response in the following.
>
> **Q1. Limited Novelty**
>
> We understand that as a unified automated visual prompting (VP) framework, the reviewer may feel that it lacks originality because it contains many visual prompting methods as special cases. However, we would like to respectfully point out several new components and designs in our AutoVP that the reviewer may overlook, as well as highlight our major contributions to VP.
>
> **[Our novel designs for VP]** AutoVP introduces novel components that have not been studied in prior arts in VP, including the **automated input scaling**, as well as **weight initialization** when using FullyMap. These additions contribute to the uniqueness of our framework, positioning AutoVP not just as a mere automated tuning tool, but as an advanced and comprehensive framework. It amalgamates a multitude of techniques to discern optimal VP configurations for datasets that exhibit diverse characteristics. More importantly, we also show that these two components are very critical to improving VP performance. For example, the optimal configurations in Table 2 suggest that different datasets prefer distinct image scales, and weight initialization with FullyMap is one of the most frequently selected output mapping methods. In doing so, our work transcends the confines of being solely a tuning tool and stands as a powerful toolset for effectively developing and deploying VP across a range of scenarios.
>
> **[Our contribution in demonstrating the advantage of VP]** It's worth noting that one of our pivotal achievements is the notable enhancement in VP performance beyond the established linear probing baseline, which had not been demonstrated in prior arts  (ILM-VP [5] and CLIP-VP [2]). In particular, our data scalability analysis in Sec. 4.1 shows that AutoVP substantially outperforms LP in the few-shot settings (10% and 1% data usage). We believe this is a significant and exciting finding, because efficient few-shot learning is exactly the motivation for prompting.
>
> **[Our contribution in providing new insights for VP]** Another noteworthy result is **our comprehensive analysis of VP on the OOD/ID dataset**, where we show that using AutoVP can significantly improve accuracy (as showcased in Figure 5(b)). This outcome highlights the capacity of AutoVP to operate with a wider prompt space when dealing with OOD datasets like SVHN and GTSRB, thereby leading to substantial accuracy gains. This aspect underscores VP's inherent adaptability to different dataset characteristics and has not been systematically studied in the existing literature.
>
> In response to this comment, we highlighted the novelty of AutoVP in the revision within the following sections. In the introduction, we have emphasized the **novel components (automated input scaling and weight initialization)** in our main contributions to provide readers with a clear impression at the outset. Additionally, we delve into a detailed discussion of the contributions made by these new modules in Section 5. Furthermore, we've incorporated the reviewer's suggestions to integrate a **segmentation task** into our revised version, available in **Appendix B.6**.

---

> ### Author Response · Authors · 2023-11-20
> **Response to Reviewer nUAn (2 of 4)**
>
> **Q2. Potential Overfitting**
>
> We are not sure what the reviewer’s meaning of “overfitting to specific tasks.” In our experiments, AutoVP is designed to automatically search for the best configuration of different downstream tasks given the **same** set of hyperparameter candidate sets, and its selection would vary according to the given datasets.
>
> We speculate the reviewer would like to learn more about whether AutoVP would “overestimate” at the initial tuning phase. Indeed, as a hyper-parameter optimization (HPO) framework, the resulting configuration among many combinations tends to be overfitted or overestimated. This phenomenon was first discovered as “the multiple induction problem” [1], also known as the “winner’s curse” in statistics. One can search all the combinations and train those models longer to avoid the sub-optimal tuning results. However, this may cost much more computation costs. Some research also suggests that if certain datasets are of good quality (well-balanced, large enough, etc.) then they would not be too sensitive to the hyper-parameter, which means the initial tuning phase is enough, and one can regard the tuning results as the optimal configuration.
>
> In our paper, we also observe a similar phenomenon: when selecting CLIP as the pre-trained model, AutoVP would overestimate the performance of FreqMap and IterMap. That is, FullyMap tends to perform poorly in the hyper-parameter tuning process, yet may achieve higher test accuracy in full training. To address this issue, we use the weight initialization on FullyMap to fix it, which we have discussed in Section 4.2.
>
> [1] Jensen and Cohen. Multiple comparisons in induction algorithms. (Machine Learning)

---

> ### Author Response · Authors · 2023-11-20
> **Response to Reviewer nUAn (3 of 4)**
>
> **Q3. Mapping Methods**
>
> This is a great suggestion! We conducted an output mapping analysis on both FreqMap and FullyMap using the DTD dataset to gain insights into how these mappings have been learned. We have updated these results in **Appendix B.10**. Figure 15 illustrates that FullyMap can be interpreted as a weighted combination of multiple source labels, where some human-readable features may exhibit similarity. For instance, in Figure 15(a), 'bumpy' shows similarities with 'custard apple’, 'disk brake’, and 'pineapple’, while in Figure 15(b), 'scaly' shares similar features with 'boa constrictor’, 'coho’, and 'common iguana’.
>
> Furthermore, when comparing FullyMap and IterMap, a significant accuracy gap is observed: FullyMap reaches 69.96%, while IterMap-1 only reaches 40.77%. However, in Figure 16, FreqMap has mapped to some classes that are indeed very close to the target. For instance, in Figure 16(b), 'braided' maps to 'knot’, 'bubbly' maps to 'bubble’, and 'cobwebbed' maps to 'spider web’. This demonstrates that a mere combination of source labels is insufficient for achieving better performance; **the weighting in the combination plays a crucial role, which is precisely what FullyMap accomplishes**.

---

> ### Author Response · Authors · 2023-11-20
> **Response to Reviewer nUAn (4 of 4)**
>
> **Q4. Segmentation and Detection**
>
> We acknowledge the need for a closer examination of various vision tasks. To address this, we integrated AutoVP into a **segmentation task** and compared its performance with linear probing on both ID and OOD datasets. The results are provided in **Appendix B.6** and table below.
>
> In our VP segmentation framework illustrated in Figure 12, a FullyMap is incorporated after the pre-trained model to enable pixel-wise classification with a custom class number. In contrast, the linear probing approach modifies the last 2D convolutional layer. Table 6 displays the outcomes.
>
> We evaluated segmentation performance using two metrics: IoU (Intersection over Union) score and pixel-wise classification accuracy. **AutoVP exhibited superior performance on both metrics in the ISIC dataset**. Additionally, segmentation examples highlighted that predictions align more accurately with the ground truth mask when the prompt space is larger (see Figure 13). However, in the ID dataset (Pets), VP performance was inferior to LP. This aligns with our findings in the classification task, where OOD datasets derived greater benefits from visual prompts.
>
> | Dataset |            LP            |          AutoVP          |
> |:-------:|:-------------------------:|:-------------------------:|
> |   Pets  | IoU : 0.83, Pixel : 90.7% | IoU : 0.77, Pixel : 86.9% |
> |   ISIC  | IoU : 0.64, Pixel : 78.1% | IoU : 0.81, Pixel : 89.5% |
>
>
> As for **detection tasks**, we are still working on the experiments. We aim to provide those results before the end of the author-reviewer rebuttal deadline.

---

> ### Author Response · Authors · 2023-11-23
> **Follow up to the Reviewer**
>
> Dear Reviewer nUAn,
>
> As the discussion period is closing soon, we'd like to follow up and see if the Reviewer has had a chance to consider our response. The additional experimental results on the detection task have been provided in our [Update Response](https://openreview.net/forum?id=wR9qVlPh0P&noteId=vayyWGrn3M). We hope our responses and new results are helpful for finalizing the review rating.
>
> Yours Sincerely,
>
> Authors

---

### Author Response · Authors · 2023-11-20
**General Response**

We thank all reviewers for your valuable feedback and helpful suggestions. With your assistance, we are committed to enhancing the clarity and conciseness of our paper. In this revised version, we have incorporated several major modifications and additions, which are outlined below and highlighted in orange in the paper.


* We've included the findings related to the **segmentation task** in AutoVP in Appendix B.6 to delve deeper into various vision tasks. The results mirror those of the classification task, showcasing greater benefits for out-of-distribution (OOD) datasets through visual prompts.


* In Appendix B.10, we extensively discuss the **mapping method**, presenting an example where FullyMap, as a weighted multi-label mapping method, outperforms other mapping methods.

* Moreover, we've incorporated the results of AutoVP on additional datasets—**SUN397, RESISC, CLEVR, and StanfordCar**—in Table 5, to enhance the comprehensiveness of our comparisons with other VP research.

We hope the revised version has addressed the reviewers’ questions and concerns. We are just one post away from answering any follow-up questions the reviewers may have, and we look forward to the reviewers’ feedback.

---

### Author Response · Authors · 2023-11-22
**A Kindly Reminder**

Dear All Reviewers,

First of all, we would like to thank you all again for your valuable time and efforts spent reviewing our paper and helping us improve it. We have also revised our paper to reflect our responses to your questions  (highlighted in $\color{orange} \text{orange}$).

As the discussion period is closing, we sincerely look forward to your feedback.

It would be very much appreciated if you could once again help review our responses and let us know if these address your concerns. We are eager to answer any further questions you might have. We strive to improve the paper consistently, and it is our pleasure to have your feedback!


Yours Sincerely,

Paper #509 Authors

---

### Author Response · Authors · 2023-11-23
**Update on Detection Task (Reviewer nUAn, PZJW, dWYk)**

Within the limited time, we are only able to explore and implement AutoVP for detection tasks from scratch. We started by spending some time integrating AutoVP to CLIP for objection detection, but we encountered some technical difficulties in inserting our modules into the CLIP pipeline for detection.

Nonetheless, as an alternative, we managed to compare AutoVP with LP on Fast R-CNN [1], one of the most standard (but relatively small) object detection models. The following table shows the results of VOCDetection and ISIC datasets. Although AutoVP does not outrun LP in this case, we believe it's due to the fact that the Fast R-CNN model is too small to allow for effective prompting. For example, the classifier is a linear layer with an output dimension of 91 (corresponding to the classes in Fast R-CNN), which is way much smaller than the output of ImageNet classifiers (1000 categories). Essentially, in a classification task, the FullyMap can potentially train around 1000 x target_class_num parameters, whereas in Fast R-CNN, there are only 91 x target_class_num of parameters available. Then we believe that the gap will be closer if the pre-trained model is larger. While we understand the detection results are preliminary due to limited time, we hope our detailed experiments on image classification, and additional experiments on image segmentation, along with this pilot study on object detection, provide sufficient evidence on the utility of AutoVP.

|    Dataset   |     LP    |   AutoVP  |
|:------------:|:---------:|:---------:|
| VOCDetection | IoU: 0.39 | IoU: 0.32 |
|     ISIC     | IoU: 0.65 | IoU: 0.51 |

[1] Ren et al. Faster R-CNN: Towards Real-Time Object Detection with Region Proposal Networks (NeurIPS 2015)

---

### Meta-Review · Area_Chair_TtGk · 2023-12-11

**Metareview:**

This paper introduces AutoVP, a framework designed to automate visual prompt (VP) design for downstream image classification tasks.
The strengths highlighted in the reviews include the paper's clarity, logical structure, and the introduction of AutoVP as a comprehensive and versatile framework for automating visual prompt (VP) design choices. The framework is praised for its modular design and extensive functionalities, offering a hyperparameter tuning tool for VP and serving as a benchmark for VP performance in image classification tasks. The paper is also commended for its well-presented findings, clear figures, and detailed statistical reports, making it accessible to readers. Additionally, the performance improvements demonstrated by models tuned with AutoVP over baseline methods are acknowledged as a positive aspect. However, the reviews highlight a need for further exploration, expanded task evaluation, and a more in-depth discussion of the framework's relationship with existing techniques. The paper's overall rating is mixed (3,5,6,8). In the rebuttal, the authors addresses the novelty issue, claiming novel proposed components that have not been studied in prior arts in VP, including the automated input scaling, which seems to be a moderately novel contribution, and also added additional empirical evaluation. Given the paper, reviews, and the rebuttal, the meta-review still thinks that the merits of this paper slightly outweigh the drawbacks and therefore recommends accept.

**Justification For Why Not Higher Score:**

The reviews highlight a need for further exploration, expanded task evaluation, and a more in-depth discussion of the framework's relationship with existing techniques. The paper's overall evaluation is varied, with some reviewers advocating for acceptance, while others lean towards rejection or find it just surpassing the acceptance threshold. After a thorough examination of the rebuttal, the meta-reviewer acknowledges the presence of novelty in the methodology but deems it not substantial enough to qualify as a breakthrough.

**Justification For Why Not Lower Score:**

The paper receives praise for its commendable clarity, well-structured presentation, and empirical results. The introduction of a versatile toolbox within the framework is applauded, emphasizing its modular design and extensive functionalities. Notably, the framework is recognized for providing a valuable hyperparameter tuning tool for visual prompts (VP) and establishing itself as a benchmark for evaluating VP performance in image classification tasks. Multiple reviewers specifically highlight the well-presented findings, clear figures, and detailed statistical reports, underscoring the overall strength of the paper.

---

### Decision · Program_Chairs · 2024-01-16

Accept (poster)